# A Unifying Perspective on Unsupervised Reinforcement Learning Algorithms

## Abstract

Many sequential decision-making domains, from robotics to language agents, are naturally multi-task on the same set of underlying dynamics. Rather than learning a policy for each task separately, unsupervised reinforcement learning (URL) algorithms pretrain without reward, then leverage that pretraining to quickly obtain performant policies for complex tasks. To this end, a wide range of algorithms have been proposed to explicitly or implicitly pretrain a representation that facilitates quickly solving some class of downstream RL problems. Examples include Goal-conditioned RL (GCRL), Mutual Information Skill Learning (MISL), Successor Feature learning (SF), among others. Amid these disparate objectives lies the open problem of selecting the appropriate representation for sequential decision-making in a particular domain. This paper brings a unifying perspective to all these distinct algorithmic frameworks that make use of the sequential data in some way to predict future outcomes. First, we show that these seemingly disjoint algorithms are, in fact, approximating a common intractable representation learning objective under differing assumptions. We illuminate how these methods make use of embeddings that compress equivalent states to tractably optimize the objective. Finally, we show that assumptions governing practical URL methods create a performance-efficiency tradeoff that can help guide algorithm selection.

## 1 Introduction

Reinforcement Learning (RL) algorithms learn complex policies by identifying the complex interplay between actions, dynamics, and reward through trial and error. While RL has seen tremendous success across different fields (Chervonyi et al., 2025; Degrave et al., 2022; Wurman et al., 2022; Guo et al., 2025; Silver et al., 2017; Fawzi et al., 2022), it still relies on using a large number of environment interactions to learn a policy, which can make it prohibitively expensive. In many settings, such as robotics, the agent needs to solve a variety of tasks, described by different reward functions, in a single environment. Learning a new policy for each new task can be prohibitively expensive. In response, Unsupervised RL (URL) offers a suite of techniques to first pretrain some useful characterization of the reward-free environment so that performant policies can be inferred efficiently for a wide variety of tasks.

Over the years, many URL algorithms (Ma et al., 2022b; Touati et al., 2023; Agarwal et al., 2025; Park et al., 2023c; Wang et al., 2024; Hu et al., 2024; Gregor et al., 2016; Machado et al., 2017a; Laskin et al., 2021) have been proposed for pretraining in the reward-free setting. Through these algorithms, structures as varied as state encoders (Rudolph et al., 2024), latent skills (Eysenbach et al., 2022a), successor representations (Dayan, 1993), or goal-conditioned policies (Agarwal et al., 2023) can be pretrained, and then utilized for rapid downstream policy inference. On the surface, these techniques appear to be optimizing very different objectives, though with the similar goal of rapid policy inference. With the proliferation of complex techniques, it can be challenging for researchers trying to apply URL to new contexts or improve upon URL techniques. Moreover, due to the varied and independent design of each algorithm, it can be very difficult to analyze the weaknesses of these algorithms when compared with others.

This work investigates a core question: Can these conceptually disparate methods be unified as variations of a single core algorithmic framework? At first glance, this may seem unlikely—these methods have significantly different loss objectives and learn different representative structures, each

based on its own intuitions and assumptions. Recent work has attempted to establish bridges between different clusters of concepts, like successor measures to representation learning (Agarwal et al., 2025; Touati & Ollivier, 2021), and goal-conditioned RL to variational skills and empowerment (Choi et al., 2021). Unlike these papers, our objective is to introduce a more comprehensive unification of a large number of URL algorithms. We aim to unify these seemingly distinct methods in two ways. First, we show that each of these objectives can be traced back to the core description of future policy-dependent state reachability, or the *successor measure*. Second, we observe that all these algorithms make assumptions and use state compression via *state feature equivalence under the successor measure* to ensure tractability. Intuitively, we hypothesize that the majority of these methods learn how the distribution of future states is affected by the policy (*successor measure*) by treating states with similar properties as equivalent (*state feature equivalence*).

A natural question arises regarding which unsupervised RL algorithms can be covered by our proposed unification. While we do not claim to entirely cover the myriad of Unsupervised RL techniques, in this work, our core contribution is to illustrate that this unified objective and structure exists in a large number of existing approaches for URL. These include, Goal-Conditioned RL (GCRL) (Ma et al., 2022b), Mutual Information Skill Discovery (MISL) (Zheng et al., 2025; Eysenbach et al., 2022a), Proto-Successor Measures(PSM) Agarwal et al. (2025), Proto-value Functions(PVF) (Mahadevan, 2005; Farebrother et al., 2023), Successor Features(SF) (Dayan, 1993; Barreto et al., 2017), Controllable Representations (Islam et al., 2023; Rudolph et al., 2024) and World Models (Hafner et al., 2020; Ding et al., 2024). These approaches are linked by the common property that they learn some quantity or structure over the environment during pretraining that reasons about future occupancy distributions. In GCRL or MISL, this happens through policy or value-derived structures; in PSM and SF, through successor measures; in PVF, through linear value functions; in controllable representations, through state embeddings; and world models use generative dynamics models. In this paper we formalize the growing body of evidence (Choi et al., 2021; Levy et al., 2023; Zheng et al., 2025; Fujimoto et al., 2025)—intuiting that since these methods learn to characterize the same information (linking actions and dynamics) to achieve the same outcome (rapid policy inference given a reward function), they are in fact fundamentally linked.

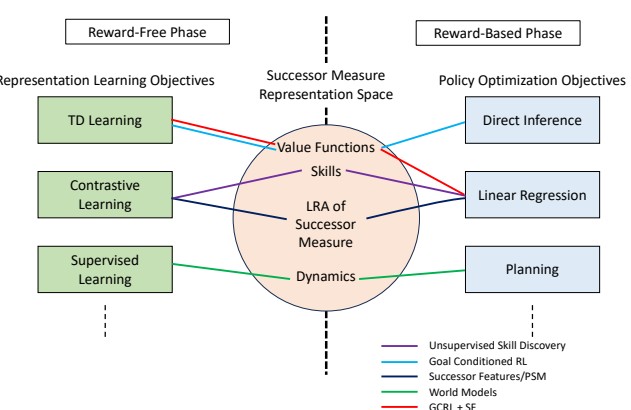

We present the unified framework in four steps. First, we introduce the unified URL objective using the notion of **successor measures** and discuss why a tractable approximation is needed for the algorithm. Second, we highlight the **assumptions and approximations** made by each of the different URL approaches towards a tractable solution of the unified objective. Third, we illustrate that to learn tractable, concise representations for successor measures, each method learns a suitable state abstraction implicitly or explicitly through a unified concept

Figure 1: **A unified perspective for Unsupervised RL**: Using the unification through successor measures, the majority of URL algorithms can be seen as a set of design choices with approximations and assumptions.

of **state equivalences**. Finally, we demonstrate the **consequences and takeaways** of these theoretical discussions as ways to construct novel algorithms and empirical evaluations of these different approaches in an environment with a large set of downstream reward functions. Our core contributions can be summarized as, (1) we identify the unified objective that all of the different methods strive for, deriving how each method can be framed as an optimization of this unified objective; (2) we identify the assumptions and approximations made by the various methods towards solving the unified objective providing a deeper theoretical understanding of each formulation; (3) we provide pathways for novel algorithm design by combining the different phases of different algorithms based on downstream application. Through our unification, we aim to inspire possible avenues of future research in unsupervised RL stemming from a better understanding of the existing methods, a careful study of their assumptions and limitations.

## 2 THE UNSUPERVISED RL PROBLEM

Before discussing the unifying objective for URL, we present the unsupervised RL problem. The URL problem is a modification of the well-known RL problem. Both problems (RL and URL) assume that an autonomous agent is operating in a Markov Decision Process (MDP) (Puterman, 2014). A Markov Decision Process is a stochastic process defined as $(\mathcal{S}, \mathcal{A}, P, r, \gamma)$ where $\mathcal{S}$ denotes the set of states; $\mathcal{A}$ denotes the set of actions; $P : \mathcal{S} \times \mathcal{A} \times \mathcal{S} \to [0, 1]$ is the transition probability function, where $P(s' \mid s, a)$ is the probability of transitioning to state $s'$ from state $s$ after taking action $a$; $r : \mathcal{S} \to \mathbb{R}$ is the reward function; and $\gamma$ is the discount factor. A policy, $\pi : S \to \Delta(A)$ is a function that outputs a distribution of actions for every state. The agent observes states and actions, but does not know the MDP's transition or reward functions, $P$ or $r$.

**The RL Problem:** Enable the agent to find the optimal policy $\pi^*$ that maximizes $J(\pi) = \mathbb{E}_\pi[\sum_t \gamma^t r(s_t)]$.

URL operates on a reward-free MDP (MDP\R) (Abbeel & Ng, 2004; Touati et al., 2023; Agarwal et al., 2025). Reward-free MDPs are defined as $(\mathcal{S}, \mathcal{A}, P, \gamma)$. Any dynamical system can be approximated using a reward-free MDP, and a near-infinite number of reward functions can be designed for an (MDP\R).

**The URL Problem:** The agent can act in a (MDP\R) to learn a representation $\mathcal{R}$ for the MDP – this could be value functions, state representations, policies, etc. The objective is to use $\mathcal{R}$ for more efficient policy learning, such as improved sample efficiency, computational efficiency, and/or wall clock time, compared to the standard reward-based RL approach of learning the policy from scratch only once the reward is given.

While prior URL work (Nair et al., 2023; Ma et al., 2022b) has used this *Reward-Based* downstream policy learning as merely an evaluation method, recent success in URL (Touati et al., 2023; Agarwal et al., 2025; Zheng et al., 2024) has been due to this *Reward-Based* phase being designed to best utilize the specific structure in $\mathcal{R}$ to lead to efficient policy learning.

Successful URL algorithms Touati et al. (2023); Agarwal et al. (2025); Zheng et al. (2024) effectively leverage the representation $\mathcal{R}$ learned through the reward-free stage at inference time, by simply using the learned representation as state (Nair et al., 2023; Ma et al., 2022b). $\mathcal{R}$ often dictates the efficiency of this downstream inference. Thus, in this work we define a **URL Algorithm**, (Laskin et al., 2021) as one which consists of an unsupervised *Reward-Free* phase to learn $\mathcal{R}$ followed by a supervised *Reward-Based* phase that efficiently uses $\mathcal{R}$ to solve the RL Problem for a wide variety of downstream tasks. The reusability of $\mathcal{R}$ and the improved efficiency of the *Reward-Based* phase makes URL Algorithms suited to settings where the agent is expected to solve a variety of tasks in the same environment.

Several frameworks and formalisms have been introduced to help solve the URL problem but the classes of approaches that we look to unify are: Goal Conditioned RL, Mutual Information Skill Learning, Successor Features, Proto-Successor Measures, Proto-Value Functions, Controllable Representations, and World Models. These seemingly disparate perspectives have a common thread among them, they learn a representation, $\mathcal{R}$ that reasons about future state occupancy distributions in some way. Detailed background and related work on each are provided in Appendix B.

## 3 SUCCESSOR MEASURE AS A UNIFYING OBJECTIVE

Each type of URL framework learns a different representation for the (MDP\$\mathcal{R}$) from the *Reward-Free* phase to allow for downstream policy inference. This raises the question: do these representations have anything in common? In this section, we argue that viewing these methods from the perspective of **Successor Measure** ($M^\pi$) estimation ties them together, bringing clarity to efficient downstream policy optimization. Mathematically, successor measure defines the measure over future states visited as $M^\pi$, $M^\pi(s, a, X) = \mathbb{E}_\pi[\sum_{t \geq 0} \gamma^t p^\pi(s_{t+1} \in X | s, a)] \ \forall X \subset \mathcal{S}$. Intuitively, it represents the discounted measure of ending up in a state $s^+ \in X$ starting from states $s$, taking an action $a$, and following the policy $\pi$ thereafter. The most common form of successor measure used is $M^\pi(s, a, s^+)$ i.e the discounted measure of ending in the state $s^+$. These methods either explicitly learn a compressed representation of successor measures or optimize a representation that allows them to implicitly use successor measure efficiently during inference. To illustrate this, we first introduce a unifying objective using successor measure. We will show that the proposed unified

objective combines these different URL formulations. Because this objective is intractable, we will introduce a tractable approximation that will provide a framework for the different URL algorithm families. In Section 4, we will discuss how a number of existing URL objectives stem from this approximation with different assumptions and present their tradeoffs.

## 3.1 THE UNIFIED FRAMEWORK

Policy optimization for any reward function can be rewritten using successor measures (Kemeny et al., 1969; Touati & Ollivier, 2021; Agarwal et al., 2025):

$$\pi^* = \arg\max_\pi \sum_{s^+} M^\pi(s, a, s^+) r(s^+). \tag{1}$$

Equation 1 indicates why successor measures form such a crucial element in URL algorithms – they provide reward-independent representations and a linear objective for policy optimization. This implies that our representations are not tied to a set of predefined tasks and that the policy optimization step can be computationally efficient based on these representations. Our proposed algorithmic framework can be divided into two phases, the **Reward-Free** or **Unsupervised Representation Learning** phase and the corresponding **Reward-Based** or **Policy Inference** phase for efficient downstream policy learning.

The *Reward-Free* phase uses interactions in the reward-free MDP to learn representations suitable for policy inference. Thus, this phase investigates the question: *how can we frontload computation for policy optimization to a pretraining stage when we don't have access to reward functions?* Successor Measure provides the answer to this question due to two key traits: 1) they are reward-free representations that can convert policy optimization into a linear objective, and 2) they characterize the notion of predicting the future distribution of an agent for any policy, which can be seen as the controllability of the agent. Then during the policy inference stage, the pretrained representation mapping from policies to a corresponding induced successor measure can be utilized to provide an optimal policy efficiently for any given reward function. In practice, based on assumptions about the distribution over downstream tasks/rewards and varying assumptions about the policy inference stage, prior URL algorithms suggest seemingly different pretraining objectives. Our proposed unified objective for unsupervised RL that covers a broad class of prior methods can be denoted as follows.

---

**Box 3.1: Unified Objective**

**Reward-Free Phase**

$$\text{Learn: } M^\pi(s, a, s^+) \qquad \forall s \in \mathcal{S} \qquad \forall a \in \mathcal{A} \qquad \forall s^+ \in \mathcal{S} \qquad \forall \pi \in \Pi \tag{2}$$

**Reward-Based Phase :**

$$\text{For a reward } r, \text{Obtain } \pi^* = \arg\max_{\pi \in \Pi} \sum_{s^+} M^\pi(s, a, s^+) r(s^+) \tag{3}$$

---

**Proposition 3.1.** *The framework presented in the Algorithm Box 3.1 is sufficient to produce optimal policies for any reward function.*

The unified objective is simple: Learn successor measures for any policy ($\Pi$ represents the class of all possible policies in the MDP), for any state-action pair. Then policy inference is simply a search using the linear product of successor measure and reward, as seen in Equation 3. However, while simple this objective is still intractable.

The main reason for intractability is that there is no way to characterize the class of all possible policies: $\Pi$. There can be $|\mathcal{A}|^{|\mathcal{S}|}$ possible deterministic policies in an MDP with finite state and action spaces, and this number can be infinite for MDPs with infinite (or continuous) states or actions. This makes characterizing a mapping from policy to the corresponding successor measures intractable. How can we perform an efficient search for $\pi \in \Pi$ during the policy inference phase from such a large non-parametric set? We introduce a tractable approximation in the next section, which we will show has connections to the different prior URL algorithms.

## 3.2 A TRACTABLE APPROXIMATION

To tackle the intractability, different URL algorithm families define parametric approximation of the policy class using latent representation $z$. Mathematically, $\Pi := \{\pi_z | z \in \mathcal{Z}\}$ with $\pi \in \Pi$ being

reduced to $z \in \mathcal{Z}$. This latent parameteric set $\mathcal{Z}$ is interpreted differently for different algorithms: these could be the set of goals (Kaelbling, 1993), set of skills (Eysenbach et al., 2018a), a set of possible linear weights for the reward span (Touati & Ollivier, 2021), or a discrete codebook (Agarwal et al., 2025). As a consequence, these frameworks have to additionally define $\mathcal{T}$ which is the set of reward functions for which their policy inference can be performed. Ideally the $\mathcal{T}$ should be the set of all reward functions but based on the approximations and assumptions on the representation space of $M^\pi$ and the space of policies $\Pi$. Due to these approximations, it may be possible that policy inference searches over a policy space that is different from $\Pi$. In the next section, we will describe both these approximations for each URL framework.

## 4 UNSUPERVISED RL OBJECTIVES UNDER THE LENS OF UNIFICATION

In this section, we pose each of the URL objectives within the same framework of estimating the successor measure. We will highlight the assumptions and compressions learned by each to produce corresponding tractable objectives that are widely used today. We will show that each of these objectives learns to represent a compact approximation of the successor measure implicitly or explicitly. These methods use this representation to either directly optimize Equation 3 or produce samples from $M^\pi$ to optimize the expectation $\mathbb{E}_{M^\pi}[r]$. We will introduce a number of cross equivalences as well that deeply connect these objectives with one another, further establishing the unification. These different methods are compared against each other based on: 1) the distribution of tasks/rewards ($\mathcal{T}$) for which they produce optimal or near-optimal policies, 2) their assumptions about the class of policy space (the latent $z$), and 3) the efficiency of their policy inference phase. The result of these equivalences is summarized in Table 1. All proofs for the theorems are included in the supplementary material.

### 4.1 GOAL-CONDITIONED REINFORCEMENT LEARNING (GCRL)

Goal-conditioned RL optimizes for a policy (and a value function) that is conditioned on the goal state $z \in \mathcal{S}$ that the agent has to reach. Mathematically, GCRL is expected to produce $V^*(s, z) = \max \mathbb{E}_\pi[\sum_t \gamma^t r_z(s_t, a_t)|s]$ (or $Q^*(s, a, z)$) where $r_z(s_t, a_t) = (1 - \gamma)p(s_{t+1} = z|s_t, a_t)$ otherwise. In its most expansive sense, the goal set is the same as the set of states with GCRL being capable of producing the value of any state conditioned on any state in the MDP.

**Under the lens of Unification**: The equivalences between GCRL and Successor measures have already been hinted at in contrastive RL Eysenbach et al. (2021) where GCRL was seen as a density estimation problem. We extend this formally here with the following assumptions.

**Assumption 4.1** (GCRL Policy Assumption). $\Pi = \{\pi_z | z \in \mathcal{S}; \pi_z \text{ is optimal to reach } z\}$.

This assumption formally defines the tractable class of policies that is considered by GCRL. Consider the next assumption on the set of tasks or rewards ($\mathcal{T}$) for which GCRL performs policy inference,

**Assumption 4.2** (GCRL Reward Assumption). $\mathcal{T} = \{(1 - \gamma)p(s_{t+1} = z|s_t, a_t) \quad \forall z \in \mathcal{Z}\}$.

With the assumptions formally defined for GCRL, we can bring GCRL into the unified objective:

**Theorem 4.3.** *With $\Pi$ and $\mathcal{T}$ defined as per Assumptions 4.1 and 4.2, GCRL learns $Q^{\pi_z}(s, a) \propto M^{\pi_z}(s, a, z)$ for $s \in \mathcal{S}, z \in \mathcal{Z}, a \in \mathcal{A}$. The optimal policy inference for reward, $r_z$ is $\pi_z$ by construction.*

### 4.2 MUTUAL INFORMATION SKILL LEARNING (MISL)

MISL objectives have been primarily used to discover skills-conditioned policies, where the skills are represented using a latent variable $Z$. While MISL approaches have large variation in their overall algorithms, the core has always been to maximize the mutual information between states and "skills" ($I(S; Z)$) or between transitions and skills ($I(S, S'; Z)$). The details of the optimization can be found in the supplementary. Since computing the mutual information exactly is intractable, MISL methods often rely on lower bounds that require training a variational distribution $q(z|s)$ (or $q(z|s, s')$) representing posterior distribution of skills which defines the reward for policy optimization conditioned on $z$.

**Under the lens of unification** We demonstrate that variational distribution $q(z|s)$ can be used to estimate successor measures (Theorem 4.6). The policy class $\Pi$ is not generally fixed in MISL, but rather emerges as a property of the objective. At convergence, the following assumption holds,

**Assumption 4.4** (MISL Policy Assumption). Let $\mathcal{Z}$ be the set of diverse skills recovered by MISL, $\Pi = \{\pi_z | z \in \mathcal{Z} \text{ i.e. } \pi_z \text{ is a skill discovered by MISL } \}$.

The set of skills discovered by MISL algorithms can be discrete (Eysenbach et al., 2018a; 2022a) or continuous (Park et al., 2023c; Zheng et al., 2025). Eysenbach et al. (2022a) makes an interesting finding that $\mathcal{Z}$ represents the set of policies optimal for some reward function and in general MISL does not recover all optimal policies.

We can define the assumption on the set of tasks considered by MISL,

**Assumption 4.5** (MISL Reward Assumption). $\mathcal{T} = \{r \mid \exists z \in \mathcal{Z} \text{ s.t. } \pi_z \in \arg\max_\pi \mathbb{E}_\pi[\sum_t \gamma^t r_t]\}$.

Finally, we can connect MISL to the unified objective using Theorem 4.6:

**Theorem 4.6.** *For $\Pi$ defined using Assumption 4.4 and $\mathcal{T}$ defined using Assumption 4.5, MISL objectives learn $M^{\pi_z}(s, s^+) = \frac{q(z|s^+,s)p(s^+|s)}{p(z)}$ for $s \in \mu$, $a \sim \pi_z(\cdot|s \sim \mu)$ and $s^+ \in \mathcal{S}$. The policy inference can be performed by searching through the space of $z \in \mathcal{Z}$ for rewards defined in $\mathcal{T}$.*

The policy inference step in the above theorem is not as simple as described, as the set of rewards $\mathcal{T}$ is not known. Prior work has used hierarchical policy inference (Eysenbach et al., 2018a) and warm starting their policy networks (Eysenbach et al., 2018a) or exploration buffers (Eysenbach et al., 2022a).

### 4.3 SUCCESSOR FEATURES (SF)

A number of prior approaches (Dayan, 1993; Barreto et al., 2017) consider a set of reward functions that are spanned by basis features (often denoted by $\phi : \mathcal{S} \to \mathbb{R}^d$) i.e. $r(s) = \sum_i \phi_i(s)w_i$ for some linear $d$-dimensional weight $w$. $\phi$ can depend on state, state-action or state-action-next state in the most general case, but for ease of exposition we restrict ourselves to state-features. For these methods, the cumulative state feature is called the successor feature, $\psi^\pi(s, a) = \mathbb{E}_\pi[\sum_t \gamma^t \phi(s_t)|s, a]$, and is used to define Q-functions (for reward $\Phi^\top w$) as $Q^\pi(s, a) = \psi^\pi(s, a)^\top w$. While several prior works (Barreto et al., 2017; Zhu et al., 2024) define the state features $\phi$ using fixed, random or Fourier features, others (Park et al., 2024; Agarwal et al., 2025) have specialized objectives that add different inductive biases to these features. There are a few methods (Touati & Ollivier, 2021; Filos et al., 2021) that have been able to jointly produce $\phi$ and $\psi$ by optimizing for $M^\pi$.

**Under the lens of unification** The connections between successor features and successor measures has already been established in prior literature (Touati et al., 2023; Agarwal et al., 2025). Here, we situate prior works in the unified framework by first posing the assumption that follows from the definition of SF:

**Assumption 4.7** (SF Reward Approximation). $\mathcal{T} = \{\mathbf{r} | \mathbf{r} = \Phi^\top z \text{ for some } z \in \mathbb{R}^d\}$.

To enable fast policy inference, a number of prior works assume an injective relationship between optimal policy and reward. Optimal policies are represented by the same latent that defines the reward function

**Assumption 4.8** (SF Policy approximation). $\Pi = \{\pi_z \mid \pi_z \text{ is optimal for the reward } \mathbf{r} = \Phi^\top z\}$.

This assumption has lead to wide success as policy inference simply boils down to linear regression to find the $z$ that fits the reward function: $z^* = \arg\min_z[(\mathbf{r} - \Phi^\top z)^2]$. This assumption also leads to suboptimalities as discussed in (Sikchi et al., 2025).

With these assumptions, we can finally write the SF in terms of the unified objective,

**Theorem 4.9.** *With $\Pi$ and $\mathcal{T}$ as defined by Assumptions 4.8 and 4.7, SF methods learn $M^{\pi_z}(s, a, s^+) = \psi(s, a, z)(\Phi^\top \Phi)^{-1}\Phi^\top, \forall s, s^+ \in \mathcal{S}$ and $a \in \mathcal{A}$. The inference on any reward function in $\mathcal{T}$ requires solving a linear regression problem, $z^* = \arg\min_z(r - \Phi^\top z)^2$.*

### 4.4 PROTO SUCCESSOR MEASURES (PSM)

Proto Successor Measure (PSM) (Agarwal et al., 2025) uses the linearity of the Bellman equations to define a decomposition of successor measure using basis vectors, $M^\pi = \phi w^\pi + b$. This parameteri-

zation makes PSM similar to successor features but the representation is simpler as $\phi$ is independent of policy $\pi$.

**Under the lens of Unification** PSM directly learns a representation for $M^\pi$ and uses these representations to infer a policy for any reward function. PSM uses a discrete codebook $z \in \mathbb{I}^+$ to parameterize the distribution of policies. The policy $\pi_z$ is given by $\text{Uniform}(z + hash(obs))$. Formally the approximation is as follows,

**Assumption 4.10** (PSM Policy Assumption). $\Pi = \{\pi_z \mid \pi_z = Uniform(z + hash(obs)), z \in [0, 2^h] \cap \mathbb{I}\}$.

PSM does not make any assumptions on the reward class and hence can produce optimal policies for $\mathcal{T} = \{$ All reward functions $\}$. The inference step requires solving a constrained linear program $\arg\max_w \phi w r$ s.t. $\phi w + b \geq 0$.

**Theorem 4.11.** *PSM learns $M^{\pi_z}(s, a, s^+) = \sum_i \phi_i(s, a, s^+) w_i^{\pi_z} + b(s, a, s^+)$ for $\pi_z \in \Pi$ as defined in Assumption 4.10. The optimal policy inference for PSM requires solving the constrained linear program $\arg\max_w \phi w r$ s.t. $\phi w + b \geq 0$.*

## 4.5 PROTO-VALUE FUNCTIONS (PVF)

Proto-Value Functions (Mahadevan & Maggioni, 2007) decompose the value function into a spectral basis, $V^\pi(s) = \phi(s)^\top w^\pi$ or $Q^\pi(s, a) = \phi(s, a)^\top w^\pi$. A number of works (Mahadevan, 2005; Farebrother et al., 2023) have extended this construction into several interesting settings. This representation looks similar to PSM, but here the value function undergoes a spectral decomposition rather than successor measures. The spectral basis has been obtained either directly using an eigen-decomposition of the graph-Laplacian (Mahadevan, 2005) or approximated as the mean error over fitting auxiliary value functions Farebrother et al. (2023); Bellemare et al. (2019).

**Under the lens of unification** Prior works Farebrother et al. (2023); Bellemare et al. (2019) have drawn connections between these representations and successor measures and the set of value functions represented by them.

**Assumption 4.12** (PVF Policy Assumption). $\Pi = \{\pi_U\}$ or a uniformly random policy.

The set of downstream tasks that can be solved by these methods is not trivial to define. Bellemare et al. (2019) describes how these spectral methods represent value functions belonging to the set $\mathcal{V} = \{V | V$ is in the convex hull of $V^{aux}\}$ where $V^{aux}$ is the set of auxiliary value functions defined by the set $V^{aux} = \{(I - \gamma P^\pi)^{-1} r_z\}$ and $r_z$ is an indicator reward $r_z = \mathbb{1}_{s=z}$. Formally, the assumption is as follows,

**Assumption 4.13** (PVF Reward Assumption). $\mathcal{T} = \{r \mid V^* \in \text{ConvexHull}(V^{aux})\}$ where $V^{aux} = \{(\mathbb{I} - \gamma P^\pi)^{-1} r_z\}$.

The following theorem connects PVF to the unified objective,

**Theorem 4.14.** *The eigenvectors used by PVFs are the same as that of $M^{\pi_U}(s, s^+)$. Therefore, PVFs learn $M^{\pi_U}(s, s^+) = \phi w$. The policy inference for a reward function in the class $\mathcal{T}$ follows from the LSPI algorithm.*

## 4.6 CONTROLLABLE REPRESENTATIONS

Controllable representation learning compresses the states to deal with only the controllable factors of the state. All of them learn state embeddings that identify what can be controlled in the state. Several prior approaches (Islam et al., 2023; Lamb et al., 2022; Levine et al., 2024; Rudolph et al., 2024) have used inverse dynamics models, $p(a|s, s')$ to model controllability. These representations learn the minimum necessary state information to recover actions, but are often insufficient to measure long term controllability. Extending these representations to multi-step requires k-step inverse dynamics models (Islam et al., 2023; Lamb et al., 2022; Levine et al., 2024) or recursive computations through Wasserstein distance (Rudolph et al., 2024).

**Under the lens of unification** These methods learn state abstractions that make them stand apart from all the other methods discussed here. But their adherence to the use of multi-step future predictability ties them back to the notion of successor measures. We start with the first assumption (4.15) that defines the setting of Exo-MDPs. The formal definition of Exo-MDPs can be found in (Efroni et al., 2022) and is also provided in the supplementary material.

**Assumption 4.15** (Exo-MDPs). It is possible to learn a mapping $\phi : \mathcal{S} \to \mathcal{X}$ with $|\mathcal{S}| > |\mathcal{X}|$ such that $\mathcal{X}$ contains all the *endogenous components*.

The inference steps of these methods also differ from those previously discussed as they do not explicitly model $M^\pi$. Rather, they use the state compression $\phi$ as a representation for downstream RL, which defines the reward functions as $\mathcal{T} = \{\text{All reward functions on } \mathcal{X}\}$.

These methods use a behavioral policy, $\pi_\beta$, to reason about multi-step controllability and learn using the successor measure based only on $\pi_\beta$, $M^{\pi_\beta}$. Methods such as Rudolph et al. (2024); Levine et al. (2024) use a uniform random policy as the behavioral policy.

**Assumption 4.16** (Controllable Representations Policy Assumption). $\Pi = \{\pi_\beta\}$.

Methods by Lamb et al. (2022); Islam et al. (2023); Levine et al. (2024) model $P(a_t|\phi(s_t), \phi(s_{t+k}))$ using a classifier $f$. They use the classifier to reason about $(s_t, s_{t+k})$ for $k \in [1, K]$. In some sense, the classifer $f$ is trying to model $\sum_{k=1}^K P(a_t|s_t, s_{t+k})$ (in case of Islam et al. (2023)) or $\sum_{k=1}^K P(a_t|s_t, s_{t+k}) = \sum_{k=1}^K f(\cdot, \cdot, k)$ (in case of Lamb et al. (2022); Levine et al. (2024). Define $M_K^\pi$ as the $K$-step undiscounted successor measure, $M_K^\pi(s, a, s^+) = \sum_{k=1}^K P(s_{t+k} = s^+|s_t, a_t)$. Consider the following theorem,

**Theorem 4.17.** *Multi-step inverse methods like Lamb et al. (2022); Islam et al. (2023); Levine et al. (2024), model $M_K^{\pi_\beta}$, $\forall s \in \mathcal{S}$, $a \in \mathcal{A}$, $s^+ \in \mathcal{S}$ as $M_K^{\pi_\beta}(s, a, s^+) = \frac{f(a|s, s^+)p^{\pi_\beta}(s^+|s)}{\pi_\beta(a|s)}$.*

On the other hand, Action-Bisimulation (Rudolph et al., 2024) uses the recursive definition of bismulation metrics to reason about an infinite horizon multi-step controllability. It can be shown through Theorem 4.18 that the state compression obtained by Action-Bisimulation is a result of equivalences predicted using successor measures,

**Theorem 4.18.** *In Action-Bisimulation (Rudolph et al., 2024), $||\phi(s_1) - \phi(s_2)|| = 0 \Leftrightarrow M^{\pi_U}(s_1, a, s^+) = M^{\pi_U}(s_2, a, s^+)$, $\forall a \in \mathcal{A}$, $s^+ \in \mathcal{S}$ where $\pi_U$ is a uniformly random policy.*

### 4.7 PLANNING WITH WORLD MODELS

Planning with World Models loosely describe a wide variety of algorithms that learn a world model to predict future trajectories given the current state, action and policy and use a variety of planning techniques to learn a policy. These could be as simple as learning single-step dynamics models (Nagabandi et al., 2019), learning latent dynamics models (Hafner et al., 2020), or learning generative models predicting several steps in the future (Ding et al., 2024). Generally, these algorithms also model the environment reward functions along with the dynamics. This discussion focuses on the unsupervised versions of these algorithms, which learn only the dynamics and use the dynamics to infer policies given the reward function.

**Under the lens of unification**: World Models are generative models for the distribution whose likelihood is defined using Successor Measures. This is investigated in $\gamma-$discounted models(Farebrother et al., 2025; Thakoor et al., 2022). We investigate the different assumptions on $\Pi$ and $\mathcal{T}$ as a consequence of single step or multi-step models. The set of policies are the ones that are covered by the dataset policy, $\pi_\beta$.

**Assumption 4.19** (World Models Policy Assumption). $\Pi = \{\pi|\pi \ll \pi_\beta\}$.

and correspondingly the reward class is defined as,

**Assumption 4.20** (World Models Task Assumption). $\mathcal{T} = \{r|\pi^*(r) \ll \pi_\beta\}$.

Assumptions 4.19 and 4.20 are for the most general case. Recent world-models (Ding et al., 2024; Janner et al., 2022) that predict several steps in the future severely restrict the policy and reward classes. With these assumptions, the unified perspective for World Models:

**Theorem 4.21.** *World Models learn the generative form of $M^\pi(s, a, s^+)$ for $\pi \in \Pi$ as defined in Assumption 4.19. The inference requires planning using samples from $M^\pi$.*

## 5 TRACTABLE OBJECTIVES REQUIRE STATE ABSTRACTIONS

In Section 3.1 we introduced the algorithmic framework 3.1 which is intractable due to the enumeration of all policies being exponential in the states. We described in Section 4 how different

| Algorithm Class | $M^\pi$ Approximation | Policy Inference | $d(\phi(s_1), \phi(s_2))$ for State Equivalences |
|---|---|---|---|
| GCRL | $Q^{\pi_z}(s,a) \propto M^{\pi_z}(s,a,z)$ | Direct for $\mathcal{T} = \{r_z(s_t, a_t) = (1-\gamma)p(s_{t+1} = z\|s_t, a_t)\}$ | $-\|\|\phi(s_1) - \phi(s_2)\|\|$ |
| MISL | $M^{\pi_z}(s, s^+) = \frac{q(z\|s^+, s)p(s^+\|)}{p(z)}$ | Search over $\mathcal{Z}$ for $\mathcal{T} = \{r \mid \pi^*(r) \in \{\pi_z\}\}$ | $D_{\mathrm{KL}}(q_\phi(z\|s_1) \;\|\| \; q_\phi(z\|s_2))$ |
| SF | $M^{\pi_z}(s, a, s^+) = \psi(s,a,z)(\Phi^\top\Phi)^{-1}\Phi^\top$ | Linear Regression for $\mathcal{T} = \{\mathbf{r}\|\mathbf{r} = \Phi^\top z \text{ for some } z \in \mathbb{R}^d\}$ | $\phi(s_1)^\top \phi(s_2)$ |
| PSM | $M^{\pi_z}(s, a, s^+) = \sum_i^d \phi_i(s, a, s^+)w_i^\pi + b(s, a, s^+)$ | Constrained LP for $\mathcal{T}$ = Any reward | $\phi(s_1)^\top \phi(s_2)$ |
| PVF | $M^{\pi_U}(s, s^+) = \phi w$ | LSPI for $\mathcal{T} = \{$ Any $r$ for which $V^* \in$ convex hull of $V^{aux}\}$ | $\phi(s_1)^\top \phi(s_2)$ |
| CR | $M_K^{\pi_\beta}(s, a, s^+) = \frac{f(z\|s, s^+)p(s^+\|s)}{\pi_\beta(a\|s)}$ | Full RL with compressed state space | $-\|\|\phi(s_1) - \phi(s_2)\|\|$ |
| World Model | $M^\pi(s, a, s+)$ is a generative model | Planning | Depends on Regularizer |

Table 1: Comparison of Unsupervised Reinforcement Learning Methods

algorithms represent successor measures for only a reduced class of policies. It is evident that there is a tradeoff in performance that depends on the size of $\Pi$. If a very large class of $\Pi$ is represented, the policy inference search is more expensive; if the class of $\Pi$ is very small, the representations are not informative enough and the optimal policy cannot be found.

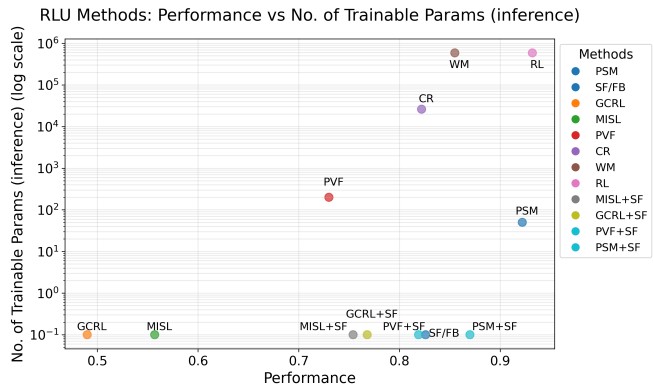

Figure 2: A **Pareto-Frontier** for URL: performance vs training cost ( the number of training parameter during the *Reward-Based* phase) between different URL formulations. These have been computed on a four room environment on a variety of tasks. Experiment Details in Appendix G.

We argue that these methods implicitly or explicitly learn state abstractions that are suitable for planning and lead to a concise form of $M^\pi$. These abstractions define state equivalences in the MDP. Formally, consider $\phi : \mathcal{S} \to \mathcal{X}$ as a state abstraction. An ideal abstraction would have $\phi(s_1) = \phi(s_2) \iff s_1 = s_2$ but this implies no compression or $|\mathcal{X}| = |\mathcal{S}|$. In practical settings, we want an abstraction that preserves the future predictability $s$. In other words, $\phi$ should be such that $M^\pi(s, a, s^+) = M^\pi(\phi(s), a, \phi(s^+))$.

Using state abstractions, state equivalences in the compressed space can be shown to follow,

**Definition 5.1.** $\phi(s_1) = \phi(s_2)$ iff $M^\pi(\phi(s_1), a, \phi(s^+)) = M^\pi(\phi(s_2), a, \phi(s^+))$.

Finally, we can define how these different URL objectives implicitly (or explicitly) define these state abstractions. For some metric $d$, $d(\phi(s_1), \phi(s_2)) \propto p(s_1 = s_2)$. The probability $p(s_1 = s_2)$ denotes the probability of the two states being equivalent. The equivalence between two states is seen through the future distributions from the two states. Specifically, $p(s_1 = s_2) \propto \mathbb{E}_{\pi \sim \Pi, a \in \mathcal{A}}\mathbf{Pr}(M^\pi(s^+\|s_1, a) \xrightarrow{d} M^\pi(s^+\|s_2, a))$. The metric $d$ is imposed on the representation space and can be independent of the underlying MDP itself. The metric $d$ is specific to the respective URL method and is mentioned in Table 1.

## 6 IMPLICATIONS OF THE UNIFICATION

The unified framework for URL sheds light on several key aspects of algorithm design.

1. **A Deeper Theoretical Understanding of URL Algorithms:** Studying each algorithm class as a series of approximations made to tractably solve the unified objective provides a novel and deeper understanding of these different formulations. This way of looking at these algorithm families directly highlight the pros and cons of these algorithm classes (Table 2).

2. **Pathway for Novel Algorithm Design:** We highlight that each algorithm learns to approximate Successor Measures during its *Reward-Free* phase and uses an inference strategy to search for the best policy in its representation class during the *Reward-Based* phase. A direct implication of this unification can be to combine the *Reward-Free* and *Reward-Based* phases of different formulations to come up with novel algorithms. Some of these have been explored in some of the recent works, CSF → MISL + SF (Zheng et al., 2024) and HILP → GCRL + SF Park et al. (2024), but a lot of such cross-combinations have not been tried yet. Our unifying framework not only provides a common ground to study these algorithms that connect different algorithm families but also paves way for novel algorithms. We provide some combinations in Figure 2. We also combine some of these methods with a policy inference by evaluating the space around the inferred optimal (which we call **Fast Finetuning**), inspired from Farebrother et al. (2025). We present some of the cross combinations in Figure 2, while the rest are provided in Table 4 and Figure 5.

   Now that we have established that these different algorithmic paradigms are striving for a tractable approximation of the unified objective of learning successor measures, we argue that future research should focus more directly on developing tractable approximations of the unified objective and better representations of Successor Measure for more efficient policy inference.

3. **Effectiveness of different algorithm families:** The goal of URL as discussed in Section 2 is to provide efficient policy inference (*Reward-Based* phase) for a large class of reward functions. The goal can be analyzed along two axes: (a) Performance over a large distribution of tasks, (b) Efficiency of policy inference. The efficiency of policy inference depends on computation of $\sum_{s^+} M^\pi(s, a, s^+) r(s^+)$ and search in $\Pi$. Figure 2 clearly shows the tradeoffs among the algorithm classes. We study the performance and efficiency on a gridworld with a wide distribution of tasks. We describe the experiment setup below and further experimental details in Appendix G.

**Experiment Setup:** We perform experiments on a four-room gridworld following prior work (Touati & Ollivier, 2021; Agarwal et al., 2025). We consider two task distributions: Goal Conditioned and RNI generated. Detailed procedure for generating these distributions is described in Appendix G. For each algorithmic paradigm, we implement their representative implementations as described in G. All these methods are tested in offline setting (no online exploration) with access to a dataset with all transitions (full coverage). The performance is measured as the number of states for which the optimal policy is predicted correctly. The efficiency is indicated as the number of trainable parameters during the policy inference as an indication compute needed during the *Reward-Based* phase. We also provide wall-clock times in Appendix G.

## 7 CONCLUSION

By pretraining models without reward, unsupervised RL holds the promise of learning good policies for a wide range of rewards. However, the abundance of recent works with disparate objectives complicates both the identification of unexplored areas and the differentiation of existing methods. In this work, we offer a framework to unify and understand some of the most popular and seemingly disparate methods. We demonstrate that each of these methods can be traced back to optimizing a form of the successor measure, and apply state equivalence to compress the underlying complexities to make this learning tractable. Through this objective, we aim to highlight connections among existing methods and, by adopting a more abstract perspective, to suggest opportunities for cross-pollination of techniques. We demonstrate that this is more than a curious observation—it informs meaningful tradeoffs between performance and efficiency and elucidates novel combinations of algorithms, some of which have been validated in literature. Thus, this work offers the promise of ushering in another wave of unsupervised RL algorithms centered around the unifying objective.

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

APPENDIX

# A    PROS AND CONS OF URL ALGORITHM FAMILIES

We continue the discussion about the pros and cons of different URL formulations from Section 6. These are a direct result of the (1) The Policy Inference method, (2) $\mathcal{T}$ or the distribution of tasks, (3) $\Pi$ or the distribution of policies for which $M^\pi$ is approximated and (4) Pretraining objective

| Algorithm Class | Pros | Cons |
|---|---|---|
| GCRL | • Inference is direct (instantaneous)
• Pretraining is easy to optimize | • $\mathcal{T}$ is restricted to only goal conditioned rewards. |
| MISL | • Allows for simultaneous training of policy and representation
• Good for online exploration | • Does not explicitly model $M^\pi$ so inference is difficult
• $\Pi$ and is often restricted. |
| SF | • Rapid inference.
• $\mathcal{T}$ is large and expressive.
• Learns $M^\pi$ for large $\Pi$. | • Requires a good choice of original features
• Requires good coverage in data, |
| PSM | • $\mathcal{T}$ is large and expressive
• Represents $M^\pi$ directly for a large $\Pi$ set. | • Representing $M^\pi$ for all policies is tough so pretraining is involved and requires coverage. |
| PVF | • The learned spaces are often intuitive to interpret. | • Policy inference is inefficient. |
| Controllable Rep. | • Representing $p(a\|s, s^+)$ is often much easier than representing $p(s^+\|s, a)$. | • Inference is involved and inefficient. |
| World Model | • Pretraining is simple. | • Policy inference requires planning which can be costly. |

Table 2: Pros and Cons of Unsupervised Reinforcement Learning Methods

# B    A DEEP DIVE INTO UNSUPERVISED RL METHODS

## B.1    GOAL CONDITIONED REINFORCEMENT LEARNING

Goal Conditioned RL refers to the class of algorithms that learn policies to reach certain goal states $g \in G$ where the set of goals is a subset of the state space $G \subseteq \mathcal{S}$. GCRL is the simplest and most common type of multi-task RL algorithms where the class of reward functions considered are simply one-hots on the goal states (notated $\mathbb{1}(s = g)$). However, even in this case a wide variety of alternative reward functions can be derived based on this including: (1-, termination, probabilistic). In Eysenbach et al. (2021) the probabilistic representation most directly captures the future state density. However, other forms have similar properties under transformations or assumptions.

A diverse set of prior works have built on the GC-MDPs (Kaelbling, 1993) to produce a large class of GCRL algorithms both in the online (Andrychowicz et al., 2017; Chuck et al., 2020; Agarwal et al., 2023; Chuck et al., 2025) and offline settings (Ma et al., 2022a; Sikchi et al., 2024). GCRL has been proposed as self-supervised learning for learning state-reaching value functions from sequential data

(Ma et al., 2022b). Several methods (Park et al., 2023c;a) use goals to define skills and use these to construct zero-shot policies (Park et al., 2023a) or for exploration (Park et al., 2023c). Goal-reaching policies can also be used as the action space for high level policies in hierarchical policy learning (Park et al., 2023a; Chuck et al., 2020; 2023), and in factored settings (Chuck, 2024; Chuck et al., 2025), where $\mathcal{Z}$ is a subset of factors, dictated by a given function $\phi : \mathcal{S} \to Z$, that selects the goal factors. Because of their simplicity, goal conditioned policies have also been applied to real world visual tasks with impressive succcess (Nair et al., 2018; Nasiriany et al., 2019).

Under the lens of unification, this diverse set of applications leverage certain assumptions about the goal space to learn the future state density, either through a representation (VIP methods) (Ghosh et al., 2018; Ma et al., 2022b) or through the value function (Choi et al., 2021). By observing this now-clarified relationship, we can not only compare the learned successor structures from GCRL to other methods that might more explicitly use successor measures like Forward Backward Representations (Touati et al., 2023) or PSM (Agarwal et al., 2025), but also utilize this to better understand the limitations of the optimal goal-reaching policy space and and uncompressed state, as compared to a parameterized space $\mathcal{Z}$, or a compressed space $\mathcal{X}$.

## B.2 MUTUAL INFORMATION SKILL LEARNING

Mutual Information Skill Learning (MISL) are a class of unsupervised RL algorithms that seeks to learn skill/option policies $\pi(a|s, z)$ that are conditioned on a latent variable $z \in Z$ representing the skills Zheng et al. (2024); Gregor et al. (2016); Park et al. (2023b); Campos et al. (2020); Laskin et al. (2022); Wang et al. (2024); Hu et al. (2024); Baumli et al. (2021). While previous MISL approaches often appear in different forms, they share a common objective of empowerment maximization, i.e. maximizing the mutual information $I(S; Z)$, where $S$ represents some environment signal derived from the state visitation, such as the final state ($s_T$) Gregor et al. (2016), any state along a trajectory ($s_t$) Eysenbach et al. (2018b), or the transition ($s_t, s_{t+1}$) Baumli et al. (2021).

Direct optimization of this mutual information objective is intractable. Instead, it can be decomposed either in the reversed or forward form:

$$I(S; Z) = H(Z) - H(Z \mid S) \quad \text{// reverse} \tag{4}$$
$$= H(S) - H(S \mid Z) \quad \text{// forward} \tag{5}$$

which gives us different ways to approximate $I(S; Z)$ via variational inference. For example, DIAYN Eysenbach et al. (2018b) utilizes the reverse decomposition:

$$I(S; Z) = \mathbb{E}_{s,z \sim p(s,z)} \left[ \log p(z \mid s) \right] - \mathbb{E}_{z \sim p(z)} \left[ \log p(z) \right] \tag{6}$$
$$\geq \mathbb{E}_{s,z \sim p(s,z)} \left[ \log q_\phi(z \mid s) \right] - \mathbb{E}_{z \sim p(z)} \left[ \log p(z) \right] \tag{7}$$

resulting in the following intrinsic reward:

$$r_{\text{int}}(s, z) = \log q_\phi(z \mid s) \tag{8}$$

Some other algorithms resort instead to the forward decomposition Laskin et al. (2022); Campos et al. (2020), resulting in objectives that encourage both conditional state predictability $q_\phi(s \mid z)$ and the state diversity $H(S)$.

Recently, variations of the original mutual information objective have been proposed, including Wasserstein dependency measure Park et al. (2023c), factorized mutual information Hu et al. (2024; 2025), and conditional mutual information based on objects or interactions Wang et al. (2024).

Specifically, METRA Park et al. (2023c), introduces a metric-aware approach to unsupervised reinforcement learning. Instead of directly maximizing mutual information between skills and states, METRA employs the Wasserstein Dependency Measure (WDM) to capture the dependency between skills and states under a distance metric $d$. In METRA, the metric $d$ is chosen to reflect the temporal distance between states, i.e., the minimum number of environment steps required to transition from one state to another. This choice of metric ensures that the learned skills are diverse in terms of their

temporal dynamics, leading to behaviors that are not only distinguishable but also cover the state space effectively.

**Empowerment-Based Skill Learning:** The empowerment objective is the mutual information between state $\mathcal{S}$ and action sequences $\vec{A}$ given initial state $\mathcal{S}^0$, for action sequences of a fixed length $I(S; \vec{A}|S_0)$. Early work in MISL pointed to a tight connection with empowerment (Eysenbach et al., 2018b), where the skill parameter $z$ can be seen as parameterizing the action sequences $\vec{A}$. Thus, adding in the initial state parameter to MISL gives the objective: $I(S; Z|S_0 = s_0)$, where MISL optimizes a variational lower bound on computing empowerment. Levy et al.; 2024) then take the MISL framework, which optimizes for the set of policies necessary to compute empowerment.

Under the lens of unification, mutual information skill learning methods represent the broad class of algorithms marrying exploration with successor measures. Through Theorem 4.6, we can view MISL methods as implicitly approximating the successor measure $M^{\pi_z}(s, a, s^+)$ by associating each skill $z$ with a distinct mode in the future state distribution. Together, the skill-conditioned policies and the variational decoder represent a structured approximation of the underlying transition dynamics. This perspective reveals that MISL implicitly encodes the dynamics of the environment through its learned latent skills, and allows for comparison against explicit successor-measure-based methods like FB (Touati et al., 2023) or PSM (Silver et al., 2017).

### B.3 SUCCESSOR FEATURES

Successor Features (Dayan, 1993; Barreto et al., 2017) are a class of multi-task RL algorithms that span rewards functions using state features as, $r = \phi w$ where $\phi$ are the state features and $w$ is the task dependent linear weight. As a consequence,

$$
\begin{aligned}
Q^\pi(s, a) &= \mathbb{E}_\pi \left[ \sum_t \gamma^t r(s_t) \right] \\
&= \mathbb{E}_\pi \left[ \sum_t \gamma^t \phi(s_t) w \right] \\
&= \mathbb{E}_\pi \left[ \sum_t \gamma^t \phi(s_t) \right] w \\
&= \psi^\pi(s, a) w
\end{aligned}
\tag{9}
$$

where, $\psi^\pi(s, a)$ is called the successor feature and is defined as, $\psi^\pi(s, a) = \mathbb{E}_\pi \left[ \sum_t \gamma^t \phi(s_t) \right]$.

Additionally, these methods align the latents of the optimal with the corresponding reward linear weights $w$ i.e. $\pi_w = \arg\max \psi^{\pi_w}(s, a) w$. This linear dependence on the optimal policy reduces policy inference to simply finding the weight $w$ corresponding to the reward function using linear regression, $w^* = \arg\min_w (\phi w - r)^2$.

A number of methods have been developed using this principle, starting from the ones using fixed, random or fourier features (Barreto et al., 2017; Zhu et al., 2024) to define the state features $\phi$ to others (Park et al., 2024; Agarwal et al., 2025) who have specialized objectives that add different inductive biases to these features.

### B.4 PROTO SUCCESSOR MEASURES

Proto Successor Measures(PSM) (Agarwal et al., 2023) uses the observation that successor measures are obey linear Bellman Equations. As a result, they can be represented using an affine set. Successor Measures are hence represented as $M^\pi(s, a, s^+) = \sum_i \phi_i(s, a, s^+) w_i^\pi + b(s, a, s^+)$ where $\phi$ are the policy independent basis functions and $b$ is the policy independent bias. $w^\pi$ is a linear weight that depends on the policy. This parameterization enables an affine representation space containing the successor measures for all policies. Unlike successor features, PSM does not directly link the policy to its corresponding reward. Given any reward function, a simple constrained Linear Program needs to be solved to obtain $w^*$.

## B.5 PROTO VALUE FUNCTIONS

Proto Value Functions refer to the class of spectral methods that linearize the value using the spectral decomposition of the graph Laplacian. They represent $V^\pi = \phi w^\pi$ where $\phi$ is independent of the policy while $w^\pi$ is a policy-dependent linear weight. Mahadevan & Maggioni (2007) approximated the graph Laplacian using a random walk operator while some (Machado et al., 2017a; Farebrother et al., 2023) have used different objectives to directly approximate the eigenfunctions. Some of these works (Farebrother et al., 2023; Bellemare et al., 2019) simply minimize the regression loss against value functions of some auxiliary tasks.

## B.6 CONTROLLABLE REPRESENTATIONS

A controllable representation is one in which only the features of the state that can change as a result of the policy are captured, and all other information is excluded. The controllable features are well described by the *endogenous* state of an Exogenous-MDP.

**Definition B.1.** (Exogenous Markov Decision Process (Efroni et al., 2022)). An exogenous-MDP (Exo-MDP) is a Block MDP where the observation $s$ can be factored into two parts $s = (x, \xi)$ where $x \in \mathcal{X}$ is the endogenous state and $\xi \in \Xi$ is the exogenous state. The transitions of the exogenous and endogenous components of the state are independent as follows: $P(s'|s,a) = P(x'|x,a)P(\xi'|\xi)$.

Methods such as (Rudolph et al., 2024; Islam et al., 2023; Efroni et al., 2022) attempt to learn an encoder $\phi : \mathcal{S} \to \mathcal{X}$ that only captures the endogenous components of the state. Notably, ACRO (Islam et al., 2023) learns the encoder $\phi$ by performing a multi-step inverse dynamics prediction between two states $k$ steps apart. The optimization is as follows,

$$\phi_\star \in \arg\max_{\phi \in \Phi} \mathbb{E}_{\substack{t \sim U(0,N) \\ k \sim U(0,K)}} \log \left( \mathbb{P} \left( a_t \mid \phi(s_t), \phi(s_{t+k}) \right) \right), \tag{10}$$

where $N$ is the maximum length of the episode and $K$ is the time horizon of interest. A small modification to this objective, as shown in Levine et al. (2024) provably extracts the full $N$-step endogenous state. In contrast, Action-Bisimulation (Rudolph et al., 2024) learns a discounted infinite-horizon representation of controllability based on a minimal single-step inverse dynamics representation. The bisimulation metric (Ferns et al., 2011) is based on the bisimulation relation Givan et al. (2003) and learns a representation to approximately obey the following relation:

$$\psi(s_i) = \psi(s_j) \tag{11}$$
$$P(\mathcal{G} \mid s_i, a) = P(\mathcal{G} \mid s_j, a) \quad \forall a \in \mathcal{A}, \forall \mathcal{G} \in \mathcal{S}_{AB}$$

where $\mathcal{S}_{AB}$ is the partition of $\mathcal{S}$ under the relation $AB$ (the set of all groups $\mathcal{G}$ of equivalent states), and

$$P(\mathcal{G} \mid s, a) = \sum_{s' \in \mathcal{G}} p(s' \mid s, a),$$

and $\psi : \mathcal{S} \to \mathcal{Z}_{ss}$ is a representation such that $p(a \mid \psi(s), \psi(s')) = p(a \mid s, s')$ for all $s, a, s'$. The single-step representation $\psi$ learns the features necessary for predicting the action taken to cause a transition. This representation is the basis of action-bisimulation because it filters out features that do not provide any signal to predict the action, i.e. anything that can be changed due to the agent's action.

While these controllable representation methods learn features that can be tied theoretically to the Unified Objective in Box 3.1, they do directly admit a policy. Instead, they provide efficient representations upon which downstream sequential decision-making tasks can be learned using RL.

## B.7 WORLD MODELS

World Models refers to a wide group of methods that learn the dynamics of the environment. They could be single-step dynamics(Nagabandi et al., 2019), latent dynamics (Hafner et al., 2020), multi-step generations (Ding et al., 2024; Janner et al., 2022), state space models (Hafner et al., 2020) or geometric models (Thakoor et al., 2022). The inference style generally requires some sort of planning depending on how ellaborate the search will be. For instance if the method learns single step dynamics (Janner et al., 2019; Nagabandi et al., 2019), often some sort of planning is needed to obtain the optimal policy. While methods like (Ding et al., 2024; Janner et al., 2022) which generate large sequences can generate multiple candidates to select from.

World Models are closely related to Successor Measures. The represent the generator functions to the otherwise density based estimation of $M^\pi$. In fact, it is the density based estimation that leads to very quick inference due quick computation of $\sum_{s^+} M^\pi(s, a, s^+) r(s^+)$. World models on the other hand compute, $\mathbb{E}_{s^+ \sim M^\pi(s^+|s,a)}[r(s^+)]$ which is time consuming and requires significantly more computation. As a result, the inference for these methods though sample efficient are not computationally efficient. Geometric Horizon Models or $\gamma$-models (Thakoor et al., 2022) are the generator functions of normalized successor measures. To learn a parametric successor measure model, simply minimize,

$$\mathbb{E}_{s \sim \rho, s^+ \sim m^\pi(\cdot|s,a)}[-\log m_\theta(s^+|s,a)] \tag{12}$$

Otherwise the density can also be obtained using samples,

$$M^\pi(s, a, s^+) = \mathbb{E}_{s \sim \mu, \pi, t \sim Geom(1-\gamma)}[\Pi_{t=1}^{T-1} p(s_t|s_{t-1}, a_{t-1})\pi(a_t|s_t)p(s_T = s^+|s_{T-1}, a_{T-1})] \tag{13}$$

## C  STATE COVERING EXPLORATION METHODS AND THEIR CONNECTIONS TO THE UNIFIED PERSPECTIVE

Exploration through intrinsic motivation has been widely studied (Burda et al., 2018b; Lee et al., 2019; Lobel et al., 2023) and closely related to unsupervised reinforcement learning. These works use intrinsic rewards to promote exploration through curiosity based objectives (Pathak et al., 2017; Burda et al., 2018b) and those maximizing state coverage (Lee et al., 2019; Agarwal et al., 2023). Often these intrinsic rewards are added to the task based rewards. These exploration strategies can also be used to collect exploratory data to be used for further training (Yarats et al., 2022). In either case, these algorithms on their own do not qualify for unsupervised RL. Through Section 2, we make it clear that URL algorithms should consists of both: an objective to abstract the knowledge about the environment from reward-free interactions and an objective for policy optimization with improved efficiency using this abstraction. While exploration methods collect "good" data, neither do they learn any representation of the environment, nor provide means of policy optimization with increased efficiency.

Nevertheless, we can connect the exploration methods aiming for state coverage to our unified perspective of successor measures. These methods estimate visitation counts for the states through various formulations (Lobel et al., 2023; Bellemare et al., 2016). Without loss of generality, lets assume $f(s)$ to be an estimation of the visitation frequency of state $s$. Also, define $\pi_{exp}^*$ be the maximally exploratory policy i.e. the policy with maximum state coverage. It can easily be shown that,

**Theorem C.1.** *State-coverage based exploration methods estimate $M^{\pi_{exp}^*}(s, a, s^+) \propto f(s^+)$ where $s \sim \mu, a \sim \pi_{exp}^*$.*

## D  SCOPE OF OUR UNIFICATION

A large number of methods have been proposed to solve the URL problem discussed in Section 2. These methods have simplified the URL problem to various other paradigms such as Goal Conditioned RL, Skill Learning, State representation learning, Successor Features etc. Naturally, a majority of these methods have used the sequential nature of data i.e. have included biases into their representative models that the data is generated from a sequential decision making process. There are methods (Shah & Kumar, 2021; Majumdar et al., 2023) that have ignored this aspect and have looked at objectives looking at each data point independently. While these methods perfectly qualify as unsupervised RL algorithms, we only unify methods that are predicting about the future predictions. These include the paradigms of Goal Conditioned RL, Mutual Information-Based-Skill Learning, Successor Features, Proto Successor Measures, Proto Value Functions, Controllable Representations and World Models.

# E  ADDITIONAL EQUIVALENCES

In this section, we shall discuss several recent methods that can be studied under the umbrella of unification to form connections across different algorithm families.

## E.1  GCRL

Approaches such as VIP (Ma et al., 2022b) and HILP (Park et al., 2024) additionally parameterize $M^{\pi_z}$ as a metric ($M^{\pi_z} \propto -\|\phi(s) - \phi(z)\|$) to provide an inductive bias for representation learning.

**Contrastive RL :** Several methods (Nair et al., 2023; Ziarko et al.) have used some form of contrastive learning to learn state representations. Prior research (Eysenbach et al., 2022b) has well connected these objectives to GCRL. The InfoNCE objective used for contrastive RL is,

$$f^* = \arg\min_f \mathbb{E}_{\{s_i,a_i,g_i\}\sim\mathcal{D}} \left[ \log \frac{e^{f(s_i,a_i,g_i)}}{\sum_{i\neq j} e^{f(s_i,a_i,g_j)}} \right] \tag{14}$$

The only difference with GCRL is that InfoNCE objectives lack the policy improvement step of RL i.e. they simply evaluate the policy in the dataset. Formally the connection with GCRL and the unified perspective of successor measures can be seen as,

**Theorem E.1.** *(Lemma 4.1 of Eysenbach et al. (2022b)) Let $\pi_\beta$ represent the dataset policy, minimizing the objective in Equation 14 learns $M^{\pi_\beta}(s,a,s^+) \propto \rho(s^+)e^{f(s,a,s^+)} \qquad \forall s, a, s^+ \in \mathcal{D}$.*

When the representation learning is combined with policy improvement, Contrastive RL can learn optimal goal-reaching policies, giving rise to similar representations as GCRL. However, the state representation learned with contrastive RL can also be independently used for the policy inference step, such as with behavior cloning (Lawson et al., 2025) or downstream RL (Schwarzer et al.). In these cases, contrastive RL serves as an alternative goal-based pretraining step to existing GCRL methods, which simply use a boostrapping objective. Some constrastive methods (Eysenbach et al., 2022b) additionally consider low rank approximations, $f(s,a,s^+) = \psi(s,a)^\top \phi(s^+)$.

## E.2  MISL

Recent work (Zheng et al., 2025) leverages the relationship between MISL objective and InfoNCE as a variational lower bound Poole et al. (2019b). An unnormalized variational lower bound can be derived for the mutual information as follows,

**Theorem E.2.** *(Zheng et al., 2025) Given a critic function, $f : \mathcal{S} \times \mathcal{S} \times \mathcal{Z} \to \mathbb{R}$, $I^\pi(S, S'; Z) \geq \mathbb{E}_{p^\pi(s,s',z)}[f(s,s',z)] - \mathbb{E}_{p^\pi(s,s')}[\log \mathbb{E}_{p(z)}[e^{f(s,s',z)}]]$ where the right hand side is the variational lower bound: $(VLB(f,\pi))$*

Theorem E.2 opens wide connections between MISL and Contrastive RL approaches based on InfoNCE objectives like (Zheng et al., 2023; Myers et al., 2024). These connections have been utilized by Zheng et al. (2025); Park et al. (2023c) to extract state-representations from MISL which are different from the traditional variational compression from $q(z|s)$ or $q(z|s, s')$.

The relationship between GCRL and MISL has been studied by prior work through the lens of variational empowerment (Choi et al., 2021). Each diverse skill, $z$, is perceived to be a goal-conditioned policy $\pi_z$ (policy conditioned to reach the goal $z$). More formally,

**Theorem E.3.** *(Choi et al., 2021) For $\mathcal{Z} = \mathcal{S}$, GCRL with $r(s|z) = -\frac{1}{\sigma^2}\|z - s\|$ is the same as solving the MISL objective with the variational distribution, $q(z|s) = \mathcal{N}(z - s, \sigma^2)$.*

## E.3  SF

The policy inference for SF involves solving a linear regression which also has a closed form solution. The Forward Backward representation (Touati & Ollivier, 2021) modifies SFs to further make the inference more efficient.

**Theorem E.4.** *If the successor measure is parameterized as,* $M^\pi(s, a, s+) = F(s, a, z)^\top B(s^+)$, *with* $B(s^+) = (\Phi^\top \Phi)^{-1} \phi^\top(s^+)$ *and* $F(s, a, z) = \psi(s, a, z)$, *the algorithm in Theorem 4.9 reduces to the FB algorithm (Touati & Ollivier, 2021). The policy inference simply becomes* $z^* = Br$.

Several SF works have been designed that have connected other forms of URL like GCRL and MISL. For instance, HILP (Park et al., 2024) uses state-features learned to be sufficient to represent goal-reaching value functions:

**Theorem E.5.** *If* $\phi = \arg\min_\phi \mathbb{E}_{s,s',g}[\ell_\tau(||\phi(s) - \phi(g)|| - \mathbb{1}_{s \neq g} - \gamma||\phi(s') - \phi(g)||)]$ *in Theorem 4.9, with* $r(s, s', z) = (\phi(s) - \phi(s'))^\top z$, *the resulting algorithm is HILP (Park et al., 2024).*

A similar connection can be drawn to recent MISL works. CSF (Zheng et al., 2025) uses an InfoNCE lower bound for the mutual information objective to learn state features which are then used to learn successor features. With Successor Features, policy inference is more efficient compared to other MISL approaches.

**Theorem E.6.** *If* $\phi = \arg\max_\phi \mathbb{E}_{p^\pi(s,s',z)}[(\phi(s) - \phi(s'))^\top z] - \mathbb{E}_{p^\pi(s,s')}[\log \mathbb{E}_{p(z)}[e^{(\phi(s) - \phi(s'))^\top z}]]$, *in Theorem 4.9, with* $r(s, s', z) = (\phi(s) - \phi(s'))^\top z$, *the resulting algorithm is CSF (Zheng et al., 2025).*

### E.4 PSM

PSM has pretty strong connections to Successor Features. Agarwal et al. (2025) had introduced the theorem,

**Theorem E.7.** *For the PSM representation* $M^\pi(s, a, s^+) = \phi(s, a, s^+)w^\pi + b(s, a, s^+)$ *and* $\phi(s, a, s^+) = \phi_\psi(s, a)^T \varphi(s^+)$, *the successor feature* $\psi^\pi(s, a) = \phi_\psi(s, a)w^\pi$ *for the state feature* $\varphi(s)^T(\mathbb{E}_\rho(\varphi\varphi^T))^{-1}$.

## F PROOFS

### F.1 PROOF OF PROPOSITION 1

**Proposition 3.1.** *The framework presented in the Algorithm Box 3.1 is sufficient to produce optimal policies for any reward function.*

*Proof.* The algorithm contained in Algorithm Box 3.1 consists of two parts:

**Pretraining:** Learning $M^\pi(s, a, s^+)$, $\forall s, a, s^+, \pi$.

**Inference:** Obtaining $\pi^*$ for the given reward function using the pretrained representations.

The pretraining step simply ensures that $M^\pi$ can be represented for any $s, a, s^+, \pi$.

As long as this is true, the question remains is if the inference step can produce optimal policies given that pretraining is true. To argue if the algorithm actually produces optimal policies for any reward function, we need to inspect inference.

The inference $Q^* = \max_\pi \sum_{s^+} M^\pi(s, a, s^+)r(s^+)$ produces a $Q^* \geq Q^\pi$ for all $\pi$. Hence for any reward function, the corresponding policy, $\max_\pi \sum_{s^+} M^\pi(s, a, s^+)r(s^+)$ produces the optimal policy as long as $M^\pi$ correctly represents successor measures for all $\pi$s.

$\square$

### F.2 PROOFS FOR SECTION 4.1

#### F.2.1 PROOF OF THEOREM 3

**Theorem 4.3.** *With* $\Pi$ *and* $\mathcal{T}$ *defined as per Assumptions 4.1 and 4.2, GCRL learns* $Q^{\pi_z}(s, a) \propto M^{\pi_z}(s, a, z)$ *for* $s \in \mathcal{S}, z \in \mathcal{Z}, a \in \mathcal{A}$. *The optimal policy inference for reward,* $r_z$ *is* $\pi_z$ *by construction.*

*Proof.* The proof follows simply from the definition of Q-function for goal conditioned RL. With reward function $r_z(s_t, a_t) = (1 - \gamma)p(s_{t+1} = z|s_t, a_t)$, the Q-function is defined as:

$$Q^{\pi_z}(s, a) = (1 - \gamma)\mathbb{E}_{\pi_z}\left[\sum_{t=0}^{\infty}[\gamma^t p(s_{t+1} = z|s_t, a_t)]\right] \tag{15}$$

$$= M^{\pi_z}(s, a, z) \tag{16}$$

$\square$

### F.3 PROOFS FOR SECTION 4.2

#### F.3.1 PROOF OF THEOREM 6

**Theorem 4.6.** *For $\Pi$ defined using Assumption 4.4 and $\mathcal{T}$ defined using Assumption 4.5, MISL objectives learn $M^{\pi_z}(s, s^+) = \frac{q(z|s^+, s)p(s^+|s)}{p(z)}$ for $s \in \mu$, $a \sim \pi_z(\cdot|s \sim \mu)$ and $s^+ \in \mathcal{S}$. The policy inference can be performed by searching through the space of $z \in \mathcal{Z}$ for rewards defined in $\mathcal{T}$.*

*Proof.* Start with the MISL conditional distribution $p(z|s^+, s)$, where $s$ is the starting state and typically omitted from MISL formulations, and $s^+$ is the current state, which is approximated by the variational distribution $q(z|s^+, s)$. Applying bayes rule gives:

$$p(z|s^+, s)p(s^+|s) = p(s^+|z, s)p(z|s) \tag{17}$$

$$\frac{p(z|s^+, s)p(s^+|s)}{p(z|s)} = p(s^+|z, s) \tag{18}$$

$$\frac{q(z|s^+, s)p(s^+|s)}{p(z)} \approx p(s^+|z, s) \tag{19}$$

$$\mathbb{E}_{\pi_z}\left[\frac{q(z|s^+, s)p(s^+|s)}{p(z)}\right] \approx M^{\pi_z}(s, s^+) \tag{20}$$

The second line replaces $p(z|s)$ with $p(z)$, because the skills in MISL are sampled independently of the starting state. $p(s^+|z, s)$ is the probability of seeing a future state $s^+$ starting from state $s$ and following a skill $z$. $p(s^+|z, s) = (1 - \gamma)\sum_{t>0} p(s_t = s^+|s, z) = M^{\pi_z}(s, s^+)$. The final transformation utilizes the fact that $z$ is the parameterization of a policy. $\square$

*Remark.* While $\frac{q(z|s^+, s)p(s^+|s)}{p(z)}$ appears to be quite messy, note that the state covering nature of MISL which arises from policies optimizing the reward $r(s^+) = \log q(z|s^+, s) + \log p(z)$ actually helps to remove the complexity. In particular, if the skills are successfully state covering from starting state $s$, then $p(s^+|s) = p(z)$, that is the likelihood of reaching a state $s^+$ from state $s$ will match the likelihood of the corresponding skill being sampled, which is just $p(z)$. This leaves: $q(z|s^+, s) \approx M^{\pi_z}(s, s^+)$, where $q$ is a variational approximation of the future state density.

#### F.3.2 ADDITIONAL EQUIVALENCES

**Theorem E.3.** *(Choi et al., 2021) For $\mathcal{Z} = \mathcal{S}$, GCRL with $r(s|z) = -\frac{1}{\sigma^2}\|z - s\|$ is the same as solving the MISL objective with the variational distribution, $q(z|s) = \mathcal{N}(z - s, \sigma^2)$.*

*Proof.* This proof can be found in Choi et al. (2021) and is summarized here. Notice that the reward for MISL policy learning is $\log q(z|s^+, s) - \log p(z)$. Assigning the space of $z$ to equal $s$, $\mathcal{Z} = \mathcal{S}$, we can then replace $q(z|s^+, s) = \log \exp(-\frac{\|z-s^+\|}{\sigma^2}) - \log(2\pi)$. Replace this value back into the reward function for GCRL, and this gives $q(z|s^+, s) = \log \exp(-\frac{\|z-s^+\|}{\sigma^2}) - \log(2\pi) + \log(2\pi) = -\frac{\|z-s^+\|}{\sigma^2}$, when $p(z)$ is a unit normal distribution. This completes the proof. $\square$

**Theorem E.2.** *(Zheng et al., 2025) Given a critic function, $f : \mathcal{S} \times \mathcal{S} \times \mathcal{Z} \to \mathbb{R}$, $I^\pi(S, S'; Z) \geq \mathbb{E}_{p^\pi(s,s',z)}[f(s, s', z)] - \mathbb{E}_{p^\pi(s,s')}[\log \mathbb{E}_{p(z)}[e^{f(s,s',z)}]]$ where the right hand side is the variational lower bound: $(VLB(f, \pi))$*

*Proof.* This proof is adapted from Zheng et al. (2025). Starting from the standard information lower bound adapted for $(S, S^+)$ and $Z$.

$$I^\pi(S, S^+; Z) \geq \mathbb{E}_\pi[\log q(z|s, s^+)] + H(Z) \tag{21}$$

$$\geq \mathbb{E}_{s, s^+ \sim \rho(\pi), z \sim p(z)}[f(s, s^+, z)] - \mathbb{E}_{s, s^+ \sim \pi}[\log \mathbb{E}_{z \sim p(z)}[\exp(f(s, s^+, z))]] \tag{22}$$

The first equation is the Barber-Agakov Inequality Barber & Agakov (2004) applied to our setting. The second plugs in an energy based variational family, where $q(z|s, s^+) = \frac{p(x) \exp(f(s, s^+, z))}{\mathbb{E}_{p(z)}[f(s, s^+, z)]}$ according to Poole et al. (2019a). Thus, the information objective of MISL is lower bounded by a successor representation on $s, s^+$ and $z$. $\square$

**Theorem F.1.** *Parameterizing $f(s, s', z)$ in Theorem E.2 as $f(s, s', z) = (\phi(s) - \phi(s'))^T z$, METRA Park et al. (2023c) is obtained as an approximation to $VLB(\phi, \pi)$.*

*Proof.* This proof is adapted from Zheng et al. (2025). Starting from the previous observation and replacing $s^+$ with $s'$ gives:

$$I^\pi(S, S^+; Z) \geq \mathbb{E}_\pi[f(s, s^+, z)] - \mathbb{E}_{s, s^+ \sim \pi}[\log \mathbb{E}_{z \sim p(z)}[\exp(f(s, s^+, z))]] \tag{23}$$

$$\geq \mathbb{E}_\pi[(\phi(s) - \phi(s'))^\top z] - \mathbb{E}_{s, s^+ \sim \pi}[\log \mathbb{E}_{z \sim p(z)}[\exp(\phi(s) - \phi(s'))^\top z]] \tag{24}$$

$$\approx \min_{\lambda \geq 0} \mathbb{E}_\pi[(\phi(s) - \phi(s'))^\top z] - \lambda(d)(1 - \mathbb{E}_{s, s' \sim \rho(\pi)}[\|\phi(s) - \phi(s')\|^2] \tag{25}$$

Where the final line replaces the log-sum-exponential term with a second order taylor approximation.

$\square$

### F.4 PROOFS FOR SECTION 4.3

#### F.4.1 PROOF OF THEOREM 9

**Theorem 4.9.** *With $\Pi$ and $\mathcal{T}$ as defined by Assumptions 4.8 and 4.7, SF methods learn $M^{\pi_z}(s, a, s^+) = \psi(s, a, z)(\Phi^\top \Phi)^{-1} \Phi^\top, \forall\ s, s^+ \in \mathcal{S}$ and $a \in \mathcal{A}$. The inference on any reward function in $\mathcal{T}$ requires solving a linear regression problem, $z^* = \arg\min_z (r - \Phi^\top z)^2$.*

*Proof.* Successor Features assume $r = \phi z$ for some linear weight $z$. This assumption directly leads to $Q^\pi(s, a) = \psi^\pi(s, a)z$ where $\psi^\pi$ is the successor feature using the state features $\phi$ (See Section B.3).

As $r = \phi z, \implies z = (\phi^T \phi)^{-1} \phi^T r$.

Substituting in $Q^{\pi_z}$ (following from Section B.3, $\pi$ is conditioned on $z$),

$$Q^{\pi_z}(s, a) = \psi(s, a, z)z$$
$$\implies Q^{\pi_z}(s, a) = \psi(s, a, z)(\phi^T \phi)^{-1} \phi^T r \tag{26}$$

Following from $Q^{\pi_z} = M^{\pi_z} r$ for all $r$, it can be shown that $M^\pi = \psi(s, a, z)(\phi^T \phi)^{-1} \phi^T$. $\square$

#### F.4.2 ADDITIONAL EQUIVALENCES

**Theorem E.4.** *If the successor measure is parameterized as, $M^\pi(s, a, s+) = F(s, a, z)^\top B(s^+)$, with $B(s^+) = (\Phi^\top \Phi)^{-1} \phi^\top(s^+)$ and $F(s, a, z) = \psi(s, a, z)$, the algorithm in Theorem 4.9 reduces to the FB algorithm (Touati & Ollivier, 2021). The policy inference simply becomes $z^* = Br$.*

*Proof.* Forward Backward representations (Touati & Ollivier, 2021) represents $M^{\pi_z}(s, a, s^+) = F(s, a, z)^\top B(s^+)$.

As a result, $Q^{\pi_z}(s, a) = \sum_{s^+} M^{\pi_z}(s, a, s^+) r_z(s^+) = \sum_{s^+} F(s, a, z)^\top B(s^+) r(s^+)$.

(Touati et al., 2023) has shown that $F(s, a, z)$ is the successor feature for the state feature $(B^\top B)^{-1} B^\top$. It can be similarly shown that, the backward network in FB is the same as $(\phi^T \phi)^{-1} \phi^T$ in the SF parameterization of $M^\pi$. $\square$

**Theorem E.5.** *If* $\phi = \arg\min_\phi \mathbb{E}_{s,s',g}[\ell_\tau(||\phi(s) - \phi(g)|| - \mathbb{1}_{s \neq g} - \gamma||\phi(s') - \phi(g)||)]$ *in Theorem 4.9, with* $r(s, s', z) = (\phi(s) - \phi(s'))^\top z$, *the resulting algorithm is HILP (Park et al., 2024).*

*Proof.* The HILP algorithm (Park et al., 2024) consists of three major steps: (1) Learning a state representation $\phi$, (2) Defining reward functions using $\phi$ and a linear weight $z$ and (3) Training $\pi_z$ to maximize $r_z$.

The first step of learning a state representation uses the following optimization,

$$\phi^* = \arg\min_\phi \mathbb{E}_{s,s',g}[\ell_\tau(||\phi(s) - \phi(g)|| - \mathbb{1}_{s \neq g} - \gamma||\phi(s') - \phi(g)||)] \tag{27}$$

The second step, defines a reward function $r(s, s', z) = \phi(s, s')z = (\phi(s) - \phi(s)')z$.

Finally, the final step requires training $\pi_z$ for corresponding $r_z$. This is achieved in practice by parameterizing the Q-function using successor features.

Hence, HILP algorithm is an SF based method with state features, $\phi$, trained using Equation 27. $\square$

**Theorem E.6.** *If* $\phi = \arg\max_\phi \mathbb{E}_{p^\pi(s,s',z)}[(\phi(s) - \phi(s'))^\top z] - \mathbb{E}_{p^\pi(s,s')}[\log \mathbb{E}_{p(z)}[e^{(\phi(s)-\phi(s'))^\top z}]]$, *in Theorem 4.9, with* $r(s, s', z) = (\phi(s) - \phi(s'))^\top z$, *the resulting algorithm is CSF (Zheng et al., 2025).*

*Proof.* Similar to the previous proof, CSF(Zheng et al., 2025) introduces a SF based algorithm that uses a MISL inspired objective to train state features, $\phi$,

$$\phi = \arg\max_\phi \mathbb{E}_{p^\pi(s,s',z)}[(\phi(s) - \phi(s'))^\top z] - \mathbb{E}_{p^\pi(s,s')}[\log \mathbb{E}_{p(z)}[e^{(\phi(s)-\phi(s'))^\top z}]] \tag{28}$$

Like HILP, CSF defines its reward function for SF as a linear span of the basis, $r(s, s', z) = \phi(s, s')z = (\phi(s) - \phi(s)')z$. $\square$

### F.5 PROOFS FOR SECTION 4.4

#### F.5.1 PROOF OF THEOREM 11

**Theorem 4.11.** *PSM learns* $M^{\pi_z}(s, a, s^+) = \sum_i \phi_i(s, a, s^+)w_i^{\pi_z} + b(s, a, s^+)$ *for* $\pi_z \in \Pi$ *as defined in Assumption 4.10. The optimal policy inference for PSM requires solving the constrained linear program* $\arg\max_w \phi wr$ *s.t.* $\phi w + b \geq 0$.

*Proof.* Proto Successor Measures (PSM) (Agarwal et al., 2025) parametrizes successor measures using an affine decomposition i.e. using basis and bias functions. Theorem 16 is a direct consequence of the parameterization. $\square$

#### F.5.2 ADDITIONAL EQUIVALENCES

**Theorem E.7.** *For the PSM representation* $M^\pi(s, a, s^+) = \phi(s, a, s^+)w^\pi + b(s, a, s^+)$ *and* $\phi(s, a, s^+) = \phi_\psi(s, a)^T \varphi(s^+)$, *the successor feature* $\psi^\pi(s, a) = \phi_\psi(s, a)w^\pi$ *for the state feature* $\varphi(s)^T(\mathbb{E}_\rho(\varphi\varphi^T))^{-1}$.

*Proof.* The proof for this theorem is adapted from Agarwal et al. (2025).

According to the PSM parameterization, $M^\pi(s, a, s^+)$ can be represented as $\phi(s, a, s^+)w^\pi$ (dropping the bias term for simplicity. It can be thought of as absorbing the bias term into the basis. If $\phi(s, a, s^+) = \phi_\psi(s, a)^T \phi_s(s^+)$, for some $\phi_\psi$ and $\phi_s$,

$$M^\pi(s, a, s^+) = \sum_i \sum_j \phi_\psi(s, a)_{ij} \phi_s(s^+)_j w_i^\pi$$

$$\implies M^\pi(s, a, s^+) = \sum_j \sum_i \phi_\psi(s, a)_{ij} w_i^\pi \phi_s(s^+)_j$$

$$\implies M^\pi(s, a, s^+) = \sum_j \phi_\psi(s, a)_j^T w^\pi \phi_s(s^+)_j$$

$$\implies M^\pi(s, a, s^+) = \sum_j \psi^\pi(s, a)_j \phi_s(s^+)_j \quad \text{(Writing } \phi_\psi(s, a)^T w^\pi \text{ as } \psi^\pi(s, a))$$

$$\implies M^\pi(s, a, s^+) = \psi^\pi(s, a)^T \phi_s(s^+)$$

From Theorem E.4, $\psi^\pi(s, a)$ is the successor feature for the basic feature $\phi_s(s)^T (\phi_s \phi_s^T)^{-1}$.

$\square$

### F.6 PROOFS FOR SECTION 4.5

#### F.6.1 PROOF OF THEOREM 14

**Theorem 4.14.** *The eigenvectors used by PVFs are the same as that of $M^{\pi_U}(s, s^+)$. Therefore, PVFs learn $M^{\pi_U}(s, s^+) = \phi w$. The policy inference for a reward function in the class $\mathcal{T}$ follows from the LSPI algorithm.*

*Proof.* PVFs learn eigenvectors for the graph laplacian given by,
$$\mathcal{L} = D - A \tag{29}$$
where $D$ is the degree matrix and A is the adjacency matrix.

The normalized graph laplacian is given by, $I - D^{-1/2} A D^{1/2}$. The random walk operator is given by,
$$L = I - T \tag{30}$$
where $T = D^{-1} A$

The Successor Representation(SR) ($\Psi^\pi$) is a quantity related to successor measures as,

$$\Psi^\pi(s, s') = \sum_{t > 0} \gamma^t \mathbb{P}(s_t = s' | s_0 = s, \pi) \tag{31}$$

Clearly, $M^\pi(s, s^+)$ is the same as $\Psi^\pi(s, s^+)$. Additionally, for a value function, $V^\pi = \Psi^\pi r = (I - \gamma P^\pi)^{-1} r$. This implies, $\Psi^\pi = (I - \gamma P^\pi)^{-1}$.

The eigen-decomposition of SR and the graph laplacians have been extensively studies by Machado et al. (2017b); Stachenfeld et al. (2014); Farebrother et al. (2023). They have shown that if $\phi$ is an eigenvector of the random walk operator ($L$), $\gamma \phi$ is the corresponding eigenvector for discounted random walk laplacian, $I - \gamma T$. And $(I - \gamma T)^{-1}$ has the corresponding eigenvector of $\gamma D^{-1/2} \phi$.

Hence, *if $\pi$ is uniform, i.e. $P^\pi = T$*, PVFs which finds the eigenvectors for the graph laplacians (random walk or normalized), also correspondingly obtain the eigenvectors for $M^{\pi_U}(s, s^+)$.

$\square$

### F.6.2 COMPARISON WITH PSM

PSM (Agarwal et al., 2025) has introduced the following theorem that compares the representative powers of PVFs compared to PSM:

**Theorem F.2.** *(Agarwal et al., 2025) Given a d-dimensional basis $\mathbf{B} : \mathbb{R}^n \to \mathbb{R}^d$, define $span\{\mathbf{B}\}$ as the span of all linear combinations of basis $\mathbf{B}$. Further define $span\{\mathbf{B}r\}$ as the span of inner products of all linear combinations of basis $\mathbf{B}$ and all possible reward functions $r$. Let $span\{\Phi^{vf}\}$ denote the space of the value functions spanned by $\Phi^{vf}$ while $\{span\{\Phi\}r\}$ denotes the space of value functions using the successor measures spanned by $\Phi$. For the same dimensionality of task (policy or reward) independent basis, $span\{\Phi^{vf}\} \subseteq \{span\{\Phi\}r\}$ for some $\Phi$.*

The theorem suggests that given the same number of dimensions, $d$, any method that spans the space of successor measures represents a larger set of value functions from the methods that span the space of value functions. We present a short adaptation of the proof from Agarwal et al. (2025).

*Proof.* We need to show that any element that belongs to the set $span\{\Phi^{vf}\}$ also belongs to the set $\{span\{\Phi\}r\}$.

Any element belonging to the set $\{span\{\Phi^{vf}\}\}$ is represented by,

$$V^{\pi}(s) = \sum_i \beta_i^{\pi} \Phi_i^{vf}(s).$$

Similarly, any element in $\{span\{\Phi\}r\}$ can be represented by,

$$V^{\pi}(s) = \sum_i w_i^{\pi} \sum_{s'} \Phi_i(s, s')r(s')$$

It is possible to show that for every element in $\{span\{\Phi^{vf}\}\}$, there exists some element in $\{span\{\Phi\}r\}$ but the reverse is not true. Only when $\Phi_i(s, s') = \sigma_i(s)\eta_i(s')$ for some $\sigma$ and $\eta$, can an element from $\{span\{\Phi\}r\}$ is present in $\{span\{\Phi^{vf}\}\}$.

$\square$

### F.7 PROOFS FOR SECTION 4.6

#### F.7.1 PROOF OF THEOREM 17

**Theorem 4.17.** *Multi-step inverse methods like Lamb et al. (2022); Islam et al. (2023); Levine et al. (2024), model $M_K^{\pi_\beta}$, $\forall s \in \mathcal{S}$, $a \in \mathcal{A}$, $s^+ \in \mathcal{S}$ as $M_K^{\pi_\beta}(s, a, s^+) = \frac{f(a|s,s^+)p^{\pi_\beta}(s^+|s)}{\pi_\beta(a|s)}$.*

*Proof.* Starting from the definition of $K$ step inverse dynamics $p(a|s, s^+)$, where $s^+$ is a state $K$ steps distant, $\pi_\beta$ is the behavior policy and $f(a, s, s^+)$ is the learned inverse dynamics, and the definition of $M_K^{\pi_\beta}(s, a, s^+) = \mathbb{E}_{\pi_\beta} p(s^{t+k} = s^+|s^t, a^t)$, we can apply bayes rule to achieve the transformations:

$$p(a|s, s^+, \pi_\beta)p(s^+|s, \pi_\beta) = p(s^+|a, s, \pi_\beta)p(a|s, \pi_\beta) \tag{32}$$

$$\frac{p(a|s, s^+, \pi_\beta)p(s^+|s, \pi_\beta)}{p(a|s, \pi_\beta)} = p(s^+|a, s, \pi_\beta) \tag{33}$$

$$\frac{p(a|s, s^+, \pi_\beta)p(s^+|s, \pi_\beta)}{\pi_\beta(a|s, \pi_\beta)} = p(s^+|a, s, \pi_\beta) \tag{34}$$

$$\frac{f(a, s, s^+)p(s^+|s, \pi_\beta)}{\pi_\beta(a|s)} \approx p(s^+|a, s) \tag{35}$$

$$\frac{f(a|s, s^+)p(s^+|s, \pi_\beta)}{\pi_\beta(a|s)} \approx M_K^{\pi_\beta}(s, a, s^+) \tag{36}$$

Notice that line 3 utilizes the fact that $p(a|s)$ in the offline distribution is the definition of the behavior policy, and line 4 uses the learned inverse dynamics to approximate the true inverse probability, where the learned inverse dynamics are learned according to $\pi_\beta$. $\square$

#### F.7.2 PROOF OF THEOREM 18

**Theorem 4.18.** *In Action-Bisimulation (Rudolph et al., 2024), $||\phi(s_1) - \phi(s_2)|| = 0 \Leftrightarrow M^{\pi_U}(s_1, a, s^+) = M^{\pi_U}(s_2, a, s^+), \forall a \in \mathcal{A}, s^+ \in \mathcal{S}$ where $\pi_U$ is a uniformly random policy.*

*Proof.* Consider the bisimulation equality for action bisimulation, where $\rho(\pi_U, s)$ is the distribution of trajectories following the uniform policy from state $s$:

$$\|\phi(s_1) - \phi(s_2)\| = \|\varphi(s_1) - \varphi(s_2)\| + \gamma \mathbb{E}_{\pi_u} \left[ \mathcal{W}(f(\cdot|s_1, a), f(\cdot|s_2, a)) \right] \tag{37}$$

$$\|\phi(s_1) - \phi(s_2)\| = \mathbb{E}_{\tau_1 \sim \rho(\pi_U, s_1), \tau_2 \sim \rho(\pi_U, s_2)} \left[ \sum_{t=0}^{\infty} \gamma^t \|\varphi(s_1^t) - \varphi(s_2^t)\|^2 \right] \tag{38}$$

The conversion between lines 1-2 simply unrolls the boostrapped wasserstein term (recall that $f : \mathcal{S} \times \mathcal{A} \to \Delta(\phi(\mathcal{S}))$, or a distribution over $\phi(s')$. Notice that the last term implies that $\|\phi(s_1) - \phi(s_2)\| = 0$ only if sum of all possible future values of $\|\varphi(s_1^t) - \varphi(s_2^t)\| = 0$, for all possible sequences of states. If this is true, since $\varphi(s)$ captures all the myopic action-relevant (and thus dynamic variability) information, $M^{\pi_U}(s_1, a, s^+) = M^{\pi_U}(s_2, a, s^+)$ for all future trajectories.

In the case where $M^{\pi_U}(s_1, a, s^+) = M^{\pi_U}(s_2, a, s^+)$, this implies also that all future distributions are the same, which means that the future trajectories match, or in other words that there is a one-to-one equivalence between $\rho(\pi_U, s_1) \equiv \rho(\pi_U, s_2) \equiv \rho(\pi_U, s_{1/2})$. Then:

$$M^{\pi_U}(s_1, a, s^+) - M^{\pi_U}(s_2, a, s^+) = 0 \qquad \Rightarrow \tag{39}$$

$$E_{\tau_1 \sim \rho(\pi_U, s_1), \tau_2 \sim \rho(\pi_U, s_2)} \left[ \sum_{t=0}^{\infty} \gamma^t \|\varphi(s_1^t) - \varphi(s_2^t)\|^2 \right] =$$

$$E_{\tau_{1/2} \sim \rho(\pi_U, s_{1/2})} \left[ \sum_{t=0}^{\infty} \gamma^t \|\varphi(s_{1/2}^t) - \varphi(s_{1/2}^t)\|^2 \right] \qquad \Rightarrow \tag{40}$$

$$\|\phi(s_1) - \phi(s_2)\| = 0 \tag{41}$$

Because the trajectories from $s_1$ and $s_2$ can be sampled equivalently. Since both $\|\phi(s_1) - \phi(s_2)\| = 0 \Rightarrow M^{\pi_U}(s_1, a, s^+) - M^{\pi_U}(s_2, a, s^+) = 0$ and $M^{\pi_U}(s_1, a, s^+) - M^{\pi_U}(s_2, a, s^+) = 0 \Rightarrow \|\phi(s_1) - \phi(s_2)\| = 0$, this means $\|\phi(s_1) - \phi(s_2)\| = 0 \iff M^{\pi_U}(s_1, a, s^+) - M^{\pi_U}(s_2, a, s^+) = 0$ $\qquad\square$

### F.7.3 PROOF OF THEOREM 21

**Theorem 4.21.** *World Models learn the generative form of $M^\pi(s, a, s^+)$ for $\pi \in \Pi$ as defined in Assumption 4.19. The inference requires planning using samples from $M^\pi$.*

*Proof.* For single step dynamics world model, $M^\pi(s, a, s^+)$ can be given as,

$$M^\pi(s, a, s^+) = \mathbb{E}_{s \sim \mu, \pi, t \sim Geom(1-\gamma)}[\Pi_{t=1}^{T-1} p(s_t|s_{t-1}, a_{t-1}) \pi(a_t|s_t) p(s_T = s^+|s_{T-1}, a_{T-1}] \tag{42}$$

$\qquad\square$

### F.8 STATE EQUIVALENCES

In Section 5, we introduced the notion that every method explicitly or through some approximations, produces state abstractions where the state space is compressed based on state equivalences. We re-introduce state-equivalences in the practical settings:

We want to learn $\phi : \mathcal{S} \to \mathcal{X}$ such that, $M^\pi(s, a, s^+) = M^\pi(\phi(s), a, \phi(s^+))$. Additionally, $\phi(s_1) = \phi(s_2)$ iff $M^\pi(\phi(s_1), a, \phi(s^+)) = M^\pi(\phi(s_2), a, \phi(s^+))$.

We mentioned that all these methods compress states based on the "distance" between the abstractions $d(\phi(s_1), \phi(s_2))$ as being proportional to $p(s_1 = s_2)$. We shall discuss the "distance" used by each of these URL algorithms:

**Goal Conditioned RL:** Goal Conditioned Value Functions have often been shown to be quasimetrics (Wang et al., 2023a) in special cases. But, in most general settings, goal conditioned value functions follow the triangle inequality (Liu et al., 2023). As a result, a number of methods (Ma et al., 2022b; Park et al., 2024) have represented value functions using L2 distances: $V(s, g) = -||\phi(s) - \phi(g)||$. These define the distances in GCRL space.

**Mutual Information Skill Learning:** MISL works compress the state representations using skills. Two states are similar if they impose the same skills. Hence the two distributions, $q(z|s_1)$ and $q(z|s_2)$ are the same if the states are equivalent (from a MISL perspective). Which means $D_{KL}(q(z|s_1)||q(z|s_2))$ represents the distance between the skill distributions for the two states $s_1$ and $s_2$.

**Successor Features:** SFs (and approximated PSM) also produce state abstractions in the form of state features. Successor measures are defined as, $M^\pi(s, a, s^+) = \sum_{t>0} p^\pi(s_t = s^+|s_0 = s, a_0 = a) = \mathbb{E}_\pi[\sum_{t>0} p(s_t = s^+|s_0 = s, a_0 = a)]$. Successor Features alternately define $M^\pi = \mathbb{E}_\pi[\sum_{t>0} \phi(s_t)^\top \phi(s^+)]$. Both these are equivalent for all $\pi$. This implies the state equivalences, $p(s_1 = s_2)$ is given by $\phi(s_1)^\top \phi(s_2)$ in case of SFs. This explains why methods (Touati et al., 2023; Touati & Ollivier, 2021) often impose orthonormality in some form in $\phi$.

**Proto Value Functions:** PVFs represent a basis for the value functions. Any two states being the same would induce the same components of the basis. Which means $\phi(s) \in \mathbb{R}^d$ will be parallel. Hence, similar to SFs, PVFs also use cosine distance, $\phi(s_1)^\top \phi(s_2)$.

**Controllable Representations:** While Islam et al. (2023); Lamb et al. (2022); Levine et al. (2024) directly optimize for state compression using the definition (by implicitly using successor measures), methods like Rudolph et al. (2024) use an L2 distance to characterize distance between two states as discussed in Theorem 4.18.

**World Models:** Latent dynamics learning (only by minimizing latent dynamics prediction error) is susceptible to collapse. Often methods add a regularizer to prevent collapse such as reconstruction, inverse kinematics model prediction, orthogonal regularization, variational losses etc. These determine the state equivalence for world models.

# G    EXPERIMENTAL EVALUATIONS

We perform experiments on a four-room gridworld following prior works (Touati & Ollivier, 2021; Agarwal et al., 2025). We build off the publically available codebase: `https://github.com/facebookresearch/controllable_agent`. The gridworld is 11x11 which is partitioned into four room with openings in the walls to traverse from one room to the other. We use the one-hot encoding of the state which makes the state 121 dimensional. There are five actions at every state, $\{stay, up, down, left, right\}$. All our experiments are offline and each algorithm has access to fully covered uniform dataset of the reward-free gridworld transitions. We choose this domain to create a wide set of tasks to analyze the performance of the various algorithms families. We have two types of tasks: *Goal-Conditioned* and *RNI generated*.

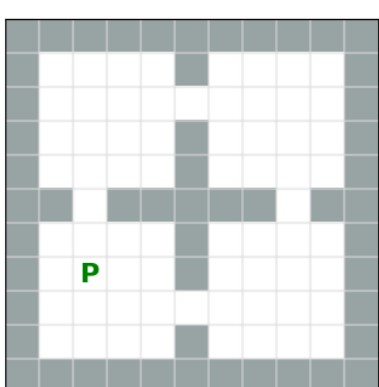

Figure 3: Example Goal Conditioned Task

**Goal Conditioned:** As the name suggests, the task consists of the agent to reach a goal. The episode *does not end* when the agent reaches the goal. In fact for both the task, the episode length is fixed to be 100 and the agents execute as if in an infinte horizon setting. (Example Figure 3).

**RNI Generated:** Prior work (Farebrother et al., 2023) used a random network indicator to generate arbitrary auxiliary tasks. The smoothness of a randomly initialize neural network does not allow absolutely random rewards but the tasks are still randomly generated and covering. For each task instantiation, we initialize a random neural network $f$ that takes in the state $s$ and assigns it a set based on a threshold.

$$r(s) = \begin{cases} 1.0, & f(s) + b_1 \geq 0 \\ -1.0, & f(s) + b_2 \leq 0 \\ 0.0, & \text{otherwise} \end{cases} \tag{43}$$

The thresholds $b_1$ and $b_2$ are obtained using expectile regression to allow $20\%$ states to have a reward of $1.0$ and $20\%$ states to have a reward $-1.0$. (Example Figure 4).

The following are the algorithm families and the representative algorithms that we implemented:

**GCRL:** We implemented a goal-conditioned double-Q learning. The inference for the goal conditioned tasks are trivial but there is no well defined way to evaluate GCRL on non-goal tasks so we do not evaluate GCRL on those tasks.

**MISL:** We implement DIAYN (Eysenbach et al., 2018b). DIAYN assumes access to a categorical distribution of skills and trains each skill using an intrinsic reward from the discriminator. We used $10$ skills (the diversity of skills were not changing as we moved to higher number of skills).

**Successor Feature:** We implement Forward-Backward Representation (Touati & Ollivier, 2021) which is the state-of-the-art SF based method.

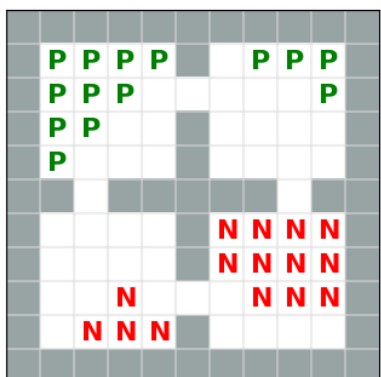

Figure 4: Example RNI Task

**PSM:** We implemented the algorithm as per the paper (Agarwal et al., 2025).

**PVF:** We implemented PVFs (Mahadevan, 2005). We parameterized the Q-value functions as a linear combination of the features.

**Controllable Representations (CR):** We implement ACRO (Islam et al., 2023) for representation learning followed by using a small mlp on top of frozen representations for policy learning.

**World Models:** We trained single step transition models and used the transition model to perform double Q learning on it for inference.

**RL:** We simple implemented a general double Q learning agent.

**Oracle:** We used tabularized Q learning to obtain optimal policy.

For each method, we measure the performance as the percent of states on which they predicted the optimal action correctly. Along with performance, we measure the number of trainable parameters during inference and wall clock time as means of measuring efficiency.

| Algorithm Class | Goal | RNI | Overall | Trainable Params | Wall Clock Time(s) |
|---|---|---|---|---|---|
| GCRL | 0.98 | 0.0 | 0.49 | 0 | 0 |
| MISL | 0.383 | 0.731 | 0.557 | 0 | 0.02 |
| SF | 0.837 | 0.814 | 0.826 | 0 | 0.00 |
| PSM | 0.906 | 0.937 | **0.922** | 50 | 40.62 |
| PVF | 0.739 | 0.720 | 0.730 | 200 | 45.86 |
| CR | 0.871 | 0.773 | 0.822 | 26112 | 55.70 |
| WM | 0.880 | 0.830 | 0.855 | 588800 | 383.21 |
| RL | 0.967 | 0.898 | **0.932** | 588800 | 70.34 |
| GCRL+SF | 0.772 | 0.764 | 0.768 | 0 | 0.03 |
| MISL+SF | 0.696 | 0.814 | 0.755 | 0 | 0.01 |
| PVF + SF | 0.798 | 0.839 | 0.819 | 0 | 0.01 |

Table 3: Performance and Efficiency of different Methods (mean over 4 seeds)

### G.1 CROSS-COMBINATIONS

For these representative methods, we construct several algorithms with the cross-combinations of the Reward-Free and Reward-Based Phases of different algorithm families. We combine different Reward-Free phases with three Reward-Based optimizations:

**SF:** Successor Features use Linear regression which has a closed form solution. So these offer one of the most efficient policy inference in terms of computation cost.

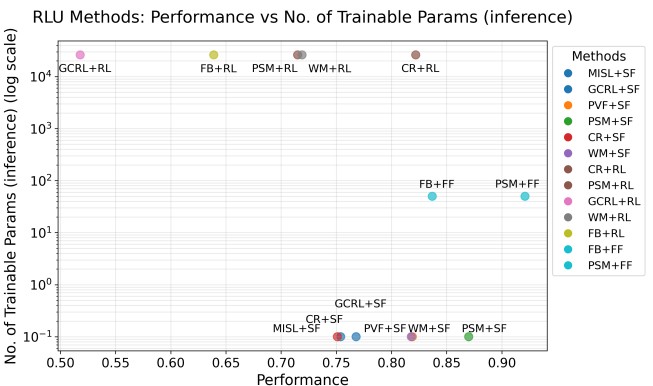

Figure 5: Performance vs training cost ( the number of training parameter during the *Reward-Based* phase) for different cross-combinations.

**RL:** An alternate could be to extract the state representations and use them as state encodings for downstream RL. While this policy inference is expensive, they dont make much assumptions about the downstream tasks.

**FF:** Fast Finetuning is an inference method where you can perturb the policies around the inferred policy and evaluate these perturbed policies using predicted successor measures. Computationally this adds some overhead to the inference methods of the representative algorithms but can potentially lead to better performance.

Some of the cross-combinations have already been tried:

**MISL+SF** We implemented CSF (Zheng et al., 2025) which is a recent work that combines MISL with SF.

**GCRL + SF** We implemented HILP (Park et al., 2024) which combines GCRL + SF.

The performance, number of trainable parameters and wall clock times are given in Table 4. We have used the dimensionality of the latent that we perturb in FF as the number of trainable parameters for it.

| Algorithm Class | Goal | RNI | Overall | Trainable Params | Wall Clock Time(s) |
|---|---|---|---|---|---|
| GCRL+SF | 0.772 | 0.764 | 0.768 | 0 | 0.03 |
| MISL+SF | 0.696 | 0.814 | 0.755 | 0 | 0.01 |
| PSM + SF | 0.912 | 0.828 | **0.870** | 0 | 0.01 |
| PVF + SF | 0.798 | 0.839 | 0.819 | 0 | 0.01 |
| CR + SF | 0.678 | 0.823 | 0.751 | 0 | 0.01 |
| WM + SF | 0.889 | 0.748 | 0.818 | 0 | 0.01 |
| GCRL+RL | 0.505 | 0.531 | 0.518 | 26112 | 49.72 |
| PSM+RL | 0.735 | 0.695 | 0.715 | 26112 | 53.49 |
| FB + RL | 0.635 | 0.646 | 0.639 | 26112 | 52.91 |
| WM + RL | 0.749 | 0.689 | 0.719 | 26112 | 53.17 |
| CR + RL | 0.871 | 0.773 | 0.822 | 26112 | 55.70 |
| FB + FF | 0.834 | 0.841 | 0.837 | 50 | 0.05 |
| PSM + FF | 0.912 | 0.929 | 0.921 | 50 | 48.61 |

Table 4: Performance and Efficiency of different Cross-Combinations (mean over 4 seeds)

**Takeaways:** From these experiments, we can infer the following:

1. The cross-combinations can improve performance and efficiency, as seen with adding SF to methods like PVF and GCRL, they might not always be successful as seen for FB + RL and PSM + RL (where there is a decrease in performance and efficiency).

2. Sometimes the representations learned can be useful for some inference (like SF) but not useful for some other (such as using them as state encoders).

3. The Fast Finetuning method adds some computation but leads to an improvement in performance.

## H ADDITIONAL UNSUPERVISED RL METHODS

While this work draws equivalences between several major classes of Unsupervised RL algorithms, we certainly do not cover all possible methods. This is not because we do not believe that these methods have relevant equivalences, but rather for time and space constraints. In this section we mention a number of additional directions that we believe share links, if not explicit reductions, to the successor measure and state equivalence abstraction. In representation learning, Bootstrap your own latent Grill et al. (2020) and Contrastive RL Eysenbach et al. (2022b) show close similarities with both action representations and successor features. Empowerment Klyubin et al. (2005); Eysenbach et al. (2018b) has long been linked to mutual information skills, while the graph Laplacian Machado et al. (2017a) and reward-free world models Ha & Schmidhuber (2018); Fujimoto et al. (2025) show close ties to spectral methods. Inverse reinforcement learning Ng et al. (2000); Ghasemipour et al. (2020) and even behavior cloning Ke et al. (2021); Brohan et al. (2023) might be seen as identifying a particular expert visitation distribution. Finally, exploration methods utilize estimates of the current state visitation distribution either through counts Burda et al. (2018b); Bellemare et al. (2016) or curiosity Wang et al. (2023b); Burda et al. (2018a); Pathak et al. (2017), and have close ties with mutual information objectives. As we can see, this work just begins a process of finding similarities and differences between existing reward-free methods. Through this work, we hope to clarify the avenues for cross-pollination and improvement in identifying the best tools when learning policies in complex environments.

## I VISUALIZING SUCCESSOR MEASURES FOR VARIOUS ALGORITHM FAMILIES

We sample random policies from the space of $\Pi$ for each of the different algorithm families and plot the successor measures $M^{\pi_z}(s_0, a_0, s^+)$ for fixed $s_0, a_0$ and $\pi_z \in \Pi$.

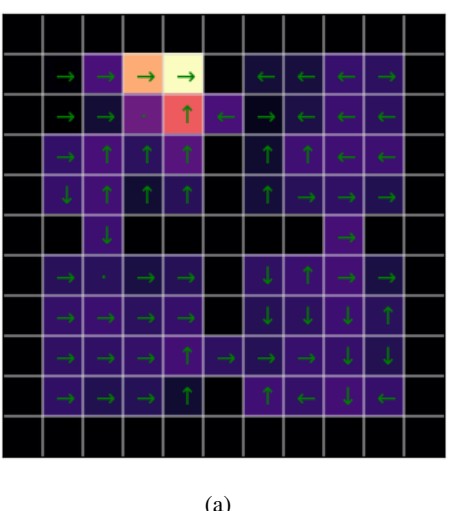
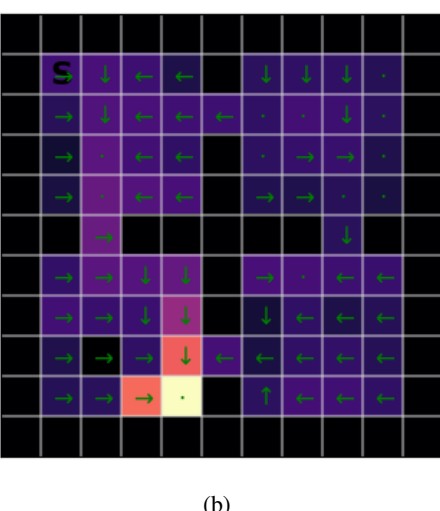

(a)                                                    (b)

Figure 6: Visualization of successor measures $M^{\pi_z}(s_0, a_0, s^+)$ for randomly sampled $z$ for FB (Successor Features).

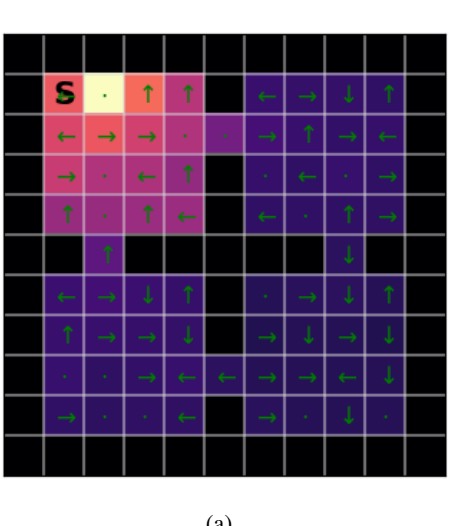 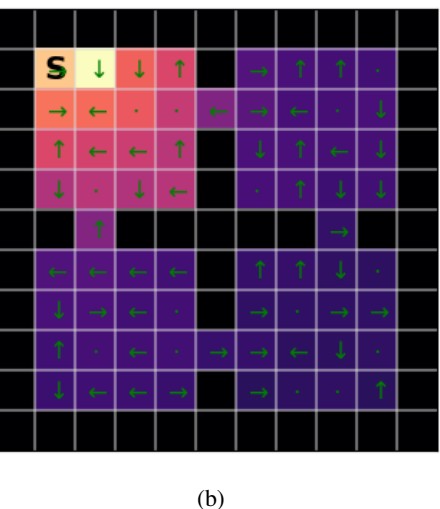

(a)            (b)

Figure 7: Visualization of successor measures $M^{\pi_z}(s_0, a_0, s^+)$ for randomly sampled $z$ for Proto Successor Measures.

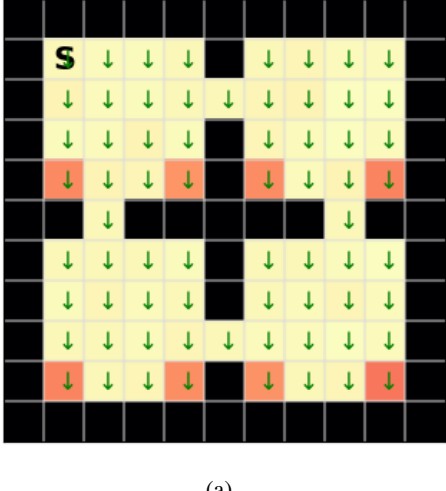 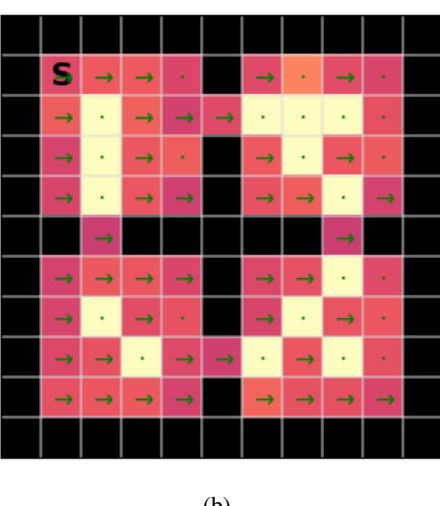

(a)            (b)

Figure 8: Visualization of successor measures $M^{\pi_z}(s_0, a_0, s^+)$ for randomly sampled $z$ for DIAYN (MISL).

