# OpenReview forum: "A Unifying Perspective on Unsupervised Reinforcement Learning Algorithms"
_ICLR.cc/2026/Conference — Submitted to ICLR 2026_

### Official Review · Reviewer_EHY5 · 2025-10-21

**Soundness:** 3
**Presentation:** 3
**Contribution:** 3
**Rating:** 8
**Confidence:** 4

**Summary:**

The paper introduces a unified framework for unsupervised reinforcement learning (URL) algorithms. The foundation of this framework is the successor measure. Within this framework, the paper proposes a single algorithm that unifies different classes of URL algorithms. The algorithm consists of two phases: a reward-free phase, in which the successor measure is learned for any policy, and a reward-based phase, in which the optimal policy is inferred using the learned successor measure and the given task reward.
The paper shows that different classes of URL algorithms (GCRL, MISL, SF, PSF, PVF, controllable representations, and world models) fall under the proposed framework. What differentiates each class is the assumptions about the policy class and the class of rewards over which the successor measure is estimated; those two assumptions influence the choice of the policy inference procedure.
The paper also suggests that different URL algorithms learn different state abstractions or notions of state equivalence to make learning tractable (either implicitly or explicitly). Finally, it highlights the trade-offs of each URL algorithm class in terms of final performance and the number of trainable parameters in the inference phase, which is proportional to the efficiency of the policy inference procedure, and discusses the implications of the proposed theoretical framework.

**Strengths:**

- The paper is clear and easy to read and follow.
- The proposed framework covers a wide range of URL algorithms.

**Weaknesses:**

- The paper does not cover URL methods that optimize state entropy; the state entropy can be connected to entropy over the successor measure, but the paper did not mention any connections.
- Some GCRL algorithms do not fall under this framework, for example, non-probabilistic GCRL methods that use some notion of distance as the reward.
- The analysis is somewhat agnostic to the settings of the pre-training (offline vs online), which might affect the policy inference phase and the set of tasks the agent assumed by the URL algorithm class.

**Questions:**

- In Theorem 4.6, if I understand correctly, the successor measure $M^{\pi_z}(s, s^+)$ is the objective when we use the posterior form of the variational distribution $q(z \mid s, s^+)$. Can you also include the successor measure when we use $q(z \mid s)$? Furthermore, would it be possible to explicitly show the link between the mutual information objective and the successor measures to make the connection clearer?
- Why are entropy/state coverage-based methods not included in the analysis? I guess most of them are considered exploration methods rather than Unsupervised Representation Learning (URL) algorithms? Can you clarify this in the paper?

---

> ### Author Response · Authors · 2025-11-19
>
> We thank the reviewer for the review. We address the concerns below:
>
> > The paper does not cover URL methods that optimize state entropy; the state entropy can be connected to entropy over the successor measure, but the paper did not mention any connections.
>
> We have added a section about exploration based methods and their connection to successor measures (Appendix C). We argue that a method that simply explores does not solve the URL problem described in Section 2. The goal of URL is to improve the efficiency of policy inference for a large wide variety of reward functions. Hence, URL algorithms must consist of both: an objective to abstract knowledge from reward-free transitions and an objective to extract policies efficiently using this abstraction. Exploration combined with representation learning (as in [1])  will lead to algorithms that fall under our definition.
>
>
> > Some GCRL algorithms do not fall under this framework, for example, non-probabilistic GCRL methods that use some notion of distance as the reward.
>
> GCRL is defined using the indicator rewards as they are the true rewards for goal conditioned RL. Using a distance based reward or a shaping reward does not necessarily lead to the optimal policies for goal conditioned RL[2].
>
>
> > The analysis is somewhat agnostic to the settings of the pre-training (offline vs online), which might affect the policy inference phase and the set of tasks the agent assumed by the URL algorithm class.
>
> The pretraining objective is guided by the assumptions made during policy inference. Having every algorithm described into these two phases allows us to construct algorithms using pretraining from one method and inference from the other (Section 6).
>
> > In Theorem 4.6, if I understand correctly, the successor measure  M^\pi_z(s, s+) is the objective when we use the posterior form of the variational distribution q(z| s, s+). Can you also include the successor measure when we use q(z|s)? Furthermore, would it be possible to explicitly show the link between the mutual information objective and the successor measures to make the connection clearer?
>
> The form $q(z|s)$ implicitly assumes an initial distribution. $q(z|s, s+)$ means the posterior distribution of skills for state $s^+$ when starting at state $s$. Using only $q(z|s)$ would mean the posterior distribution of skills for state $s$ which would implicitly assume that the agent started from some distribution (could be uniform over the state space). The quantity then learnt would be a visitation distribution which is the expected successor measure: $d^\pi(s^+) = \mathbb{E}_{s \in \mu, a \in \pi(a|s)}[M^\pi(s, a, s^+)]$.
>
> > Why are entropy/state coverage-based methods not included in the analysis? I guess most of them are considered exploration methods rather than Unsupervised Representation Learning (URL) algorithms? Can you clarify this in the paper?
>
> We have included sections C and D in the appendix discussing exploration methods and the scope of the unification. Simply exploring is not enough for a method to be solving the URL problem. Exploration combined with representation learning will lead to algorithms that fall under our definition.
>
> [1]: Liu and Abbeel, Behavior From the Void: Unsupervised Active Pre-Training, 2021
>
> [2]: A. Ng, Daishi Harada, and Stuart J. Russell. Policy invariance under reward transformations:
> Theory and application to reward shaping. In International Conference on Machine
> Learning, 1999

---

> > ### Comment · Reviewer_EHY5 · 2025-11-24
> > **Response to Authors**
> >
> > Thank you for the clarifications.
> >
> > > Exploration combined with representation learning (as in [1]) will lead to algorithms that fall under our definition.
> >
> > In this case, I suggest making a distinction in Appendix C,(1) exploration, combined with representation learning methods such as [1] and [3] falls under your framework, while other methods that combine the intrinsic reward with task reward do fall under the framework. If I understood fine-tuning is considered a precedure for policy inference right? as long as it is more efficient than on the task directly without the pre-training/reward-free phase.
> >
> >
> > [3] Yarats, Denis et al. “Reinforcement Learning with Prototypical Representations.” International Conference on Machine Learning (2021).
> >
> > Best.

---

> > > ### Author Response · Authors · 2025-11-28
> > >
> > > We thank the reviewer for the great suggestion. We will be making this distinction in our discussion on exploration in the revised version.

---

### Official Review · Reviewer_mkCJ · 2025-10-30

**Soundness:** 3
**Presentation:** 3
**Contribution:** 2
**Rating:** 2
**Confidence:** 4

**Summary:**

The paper studies aspects of the unsupervised RL problem, in which several samples are collected from a reward-free environment in order to improve the performance on a variety of downstream tasks (with their respective rewards) later on. The paper aims to unify a family of approaches, including goal-conditioned RL, skill discovery with mutual information, successor features, under the same umbrella. First, the paper casts each of these methods as an approximation of the (intractable) objective of learning the successor measure (discounted probability of reaching a state in the future) for any policy. Then, the paper characterises the approximation through the lens of state aggregations.

**Strengths:**

- The paper provides a unifying framework for a group of unsupervised RL approaches, an area of research that can definitely benefit from some clarity given the (sometimes confusing) variety of objectives and methods in the literature;
- The paper is original to the best of my knowledge;
- The paper provides specific results on the approximation of the unifying objective given by the different approaches;
- The paper proposes an ideal (intractable) characterization of unsupervised RL as the problem of learning the successor measure for any policy.

**Weaknesses:**

- The paper provides some interesting insights, but, in my opinion, the case of how future research can benefit from the presented findings is rather weak (Sec. 6);
- The title and abstract are perhaps too far reaching in their claim. The paper deals with a sizeable set of unsupervised RL approaches, but doesn't cover the entirety of previous literature, including some notable areas like policy pre-training;
- The paper only provides an informal definition of URL, as a setting composed by a reward-free phase to learn a representation to be used in a subsequent reward-based phase, but does not clarify the learning objective rigorously (e.g., how is the efficiency of the reward-based phase measured?)
- The paper seems to violate the mandatory citation style of ICLR.

**Evaluation**

In my opinion, the premise of the paper, coarsely, that the unsupervised RL problem is equivalent to learning approximate successor features is of great interest (if verified). However, I am providing a negative evaluation based on: (1) I am concerned the unsupervised RL problem definition is not rigorous enough, which makes the whole paper stands on shaky foundations, (2) I am not sure how the proposed unification brings value to the community and will help future research. I am reporting detailed comments below.

**Questions:**

**(Major) What does unsupervised RL means**

The paper characterize an URL problem as a learning problem in which there is a reward-free phase (i) devoted to learn a representation to be used in a reward-based phase (ii) to infer the optimal policy (same MDP in both phases). I think the definition of the two phases is not formal enough.

(i) reward-free phase

It's important to clarify what are the constraints of this phase.
- Are we allowed to take infinite samples? If so, one would argue that learning the transitions of the MDP is enough to solve whatever RL problem afterwards.
- Does the representation need to be succinct? Perhaps the transition model is not compact enough, but it's hard to say which direction to take without a formal specification of the objective in the subsequent phase.

(ii) reward-based phase

The stated objective "improved sample efficiency, computational efficiency, and/or wall clock time, compared to the standard reward-based RL approach of learning the policy from scratch only once the reward is given" is far from clear. Especially,
- What does *improved* means? I guess if sample efficiency goes from $N$ without any representation to $N - 1$ with the representation, we wouldn't be satisfied with the reward-free phase.
- What is standard RL here? Are we comparing with the theoretical lower bound or a specific RL method?
- Sample efficiency, computational efficiency, wall clock time... this is arguably too broad to be able to say anything interesting. I would suggest to narrow the definition. Theoretical works are typically much more formal in setting the objective, e.g., Xie et al., Policy Finetuning: Bridging Sample-Efficient Offline and Online Reinforcement Learning, 2021.

I concede that this characterization escaped the unsupervised RL community for a while, but I believe it is necessary to prove the premise of the paper.

**(Major) Main hypothesis**

The paper is based on the following

*we hypothesize that these methods [URL] learn how the distribution of future states is affected by the policy (successor measure) by treating states with similar properties as equivalent (state feature equivalence)*

Unfortunately, I am skeptical that this is true. There's a sizeable literature in URL (e.g., Liu and Abbeel, Behavior From the Void: Unsupervised Active Pre-Training, 2021) in which the efficiency of the policy inference in the reward-based phase is improved by using the reward-free phase to learn a policy that collects "good" data. I don't see how we can prove that an implicit successor measure is learned here.

My take is that saying that this analysis "unifies URL" is too bold of a claim. I think it shall be formally clarified which subset of URL methods is unified here.

**(Minor) The value of unification**

It is important to note that unifying various algorithms into a unique framework does not necessarily meet the standard for publication at a top ML conference. While the authors provide some discussion on how the unification will help future research (Sec. 6), I think the main points are mostly speculative.

**(Minor) Formal result**

Other than showing that a bunch of different methods can be understood as approximating a successor measure, it would be interesting to see whether it can be proved that any reward-free algorithm that improves the efficiency of policy inference with reward by a certain amount has to implicitly learn successor measures. See the paper Richens et al., General agents need world models, 2025 for potential inspiration.

---

> ### Author Response · Authors · 2025-11-19
>
> We thank the reviewer for the assessment of our paper. We address the questions and weaknesses raised by the reviewer below:
>
> > The paper provides some interesting insights, but, in my opinion, the case of how future research can benefit from the presented findings is rather weak.
>
> Scientifically speaking, the value of unifying theories ought to be self-evident.  They provide terminology and understanding - conceptual frameworks - that can clarify the strengths and weaknesses of various approaches and point out opportunities for new algorithms. Our work provides a theoretical foundation for a number of unsupervised RL algorithms and this clarification can have a number of implications as discussed in Section 6. We have also highlighted (corroborated by some experiments)  in Section 6 how these observations can lead to newer algorithms by combining principles from the two paradigms. We are working on experiments to add additional such combinations and will update the paper with these results soon.
>
> > The title and abstract are perhaps too far reaching in their claim. The paper deals with a sizeable set of unsupervised RL approaches, but doesn't cover the entirety of previous literature, including some notable areas like policy pre-training;
>
> We do not claim in the paper that we cover the entirety of previous literature. We are clear in our introduction that “While we do not claim to entirely cover the myriad of Unsupervised RL techniques, in this work, our core contribution is to illustrate that this unified objective and structure exists in a large number of existing approaches for URL.” In our abstract too, we state that “This paper brings a unifying perspective to all these distinct algorithmic frameworks that make use of the sequential data in some way to predict future outcomes.” which is our premise that we are unifying all the diverse classes of URL methods that are making use of predictions over future outcomes using a reward-free objective. We have added a discussion about the scope of the unification in the Appendix. We are happy to add further clarification if the reviewer has specific concerns not covered in the paper. We are also happy to modify the abstract and the title in the revised version to make sure we are not overclaiming.
>
> > The paper only provides an informal definition of URL, as a setting composed by a reward-free phase to learn a representation to be used in a subsequent reward-based phase, but does not clarify the learning objective rigorously (e.g., how is the efficiency of the reward-based phase measured?)
>
> We want to point out that our objective is to describe an existing group of methods that have claimed to be unsupervised RL, rather than prescribing a definition of the field.
> We provide the description of unsupervised RL as a reward free pretraining phase followed by amortized policy inference leveraging the representations obtained during pretraining as a broad description of the kinds of algorithms that have been self-proposed as unsupervised RL algorithms.
> We have introduced a clean description of the URL Problem that is general enough to include most of the algorithms that were proposed as unsupervised RL algorithms.
>
> The efficiency can be measured in a number of ways, such as, worst case sample complexity, amortized sample complexity, computational complexity, etc. All these quantities point towards the efficiency of policy optimization during reward based phases. The different unsupervised algorithms have provided improvements in the efficiency on different axes of sample complexity or computation complexity. There is no one axis of improvement that has been studied across different methods. For instance, learning a transition model during pretraining can use a reward-based phase that is very sample efficient but might not be computationally efficient.
>
> In our experiments, we take an approach that allows us to compare different methods under a practically relevant metric. Sample complexity is not meaningful in offline RL( the setting of our experiments), the number of gradient steps/computation is dependent on hyperparameters. The number of trainable parameters provides us a valuable way to quantify the learning complexity while being practically relevant.
>
> > The paper seems to violate the mandatory citation style of ICLR.
>
> We have fixed it. Thank you for pointing this out.

---

> > ### Author Response · Authors · 2025-11-19
> >
> > > Are we allowed to take infinite samples? If so, one would argue that learning the transitions of the MDP is enough to solve whatever RL problem afterwards.
> >
> > There are no restrictions on the amount of samples taken during the Reward-Free phase because the goal is to amortize the computation of policy optimization using the representations learnt in this phase. While the samples during the Reward-Free phase are free, they are not infinite. Due to the practical restrictions of compute and memory, any practical algorithm will resort to some form of compression and abstraction which is evident from the representations learnt by them and pointed out in our definition.
> >
> > Learning all the transitions of the MDP would indeed be an unsupervised RL algorithm and also within our unification. Our discussion on world models is exactly about learning a model for the transitions of the MDP. While this paradigm would be very efficient if the design requires improvement of the Reward-Based phase to be in terms of sample complexity, there are other algorithms that provide computationally better Reward-Based phases (highlighted through our experiments in Figure 1 and Table 2).
> >
> > > Does the representation need to be succinct? Perhaps the transition model is not compact enough, but it's hard to say which direction to take without a formal specification of the objective in the subsequent phase.
> >
> > Our paper does not give a prescription if the representations should be compact, we only seek to situate different methods (even with a large representation) with the assumptions they make.
> >
> > > We are addressing the concerns regarding the reward-based phase here:
> >
> > We would like to reiterate that we provide a description of the URL that is general enough to include a large number of self-proposed unsupervised RL algorithms. We describe the goal of URL to amortize the computation of RL algorithms into two phases, leading to improved efficiency of the Reward-Based phase. This improvement depends on how successful an underlying algorithm has been at amortizing the computation. An algorithm that does not improve the efficiency significantly will also be a URL algorithm but arguably not a good one. We would like the reviewer to refer to the response above about the various ways to measure this improvement in efficiency.
> >
> > We would like to point out that we do not wish to prescribe any specific improvement in policy extraction as being better than the others. We have pointed out in the paper (Table 1) that while Goal Conditioned RL could arguably have the best possible efficiency of the Reward-Based phase in terms of computation and sample complexity, and they provide an easier (more stable and scalable) objective for the Reward-Based phase, they only cover a small set of reward functions.
> >
> > > What is standard RL here? Are we comparing with the theoretical lower bound or a specific RL method?
> >
> > By standard RL we mean any general RL algorithm i.e. the theoretical lower bound.
> >
> > > There's a sizeable literature in URL (e.g., Liu and Abbeel, Behavior From the Void: Unsupervised Active Pre-Training, 2021) in which the efficiency of the policy inference in the reward-based phase is improved by using the reward-free phase to learn a policy that collects "good" data. I don't see how we can prove that an implicit successor measure is learned here.
> >
> > We have added a discussion on exploration methods in Appendix C. We would argue that only collecting “good” data is not enough but URL algorithms must consist of both: an objective to abstract the knowledge from reward-free interactions and an objective to utilize this abstraction to make the policy extraction efficient. The work (Liu and Abbeel, Behavior From the Void: Unsupervised Active Pre-Training, 2021) not only explores but learns a representation that is then used by the Reward-Free phase. So this work is also solving the URL problem that we have described. In Appendix C, we have also included how several of these exploration methods can be connected to successor measures as they often model the visitation counts for the exploratory policy.
> >
> > > My take is that saying that this analysis "unifies URL" is too bold of a claim. I think it shall be formally clarified which subset of URL methods is unified here.
> >
> > We do not propose that our paper unifies RL. We have already specified the subset of URL methods that we unify in our paper including in our abstract and introduction (Please refer to the response above). We have also included a section in the Appendix (Appendix D). We are happy to provide additional clarifications to make sure we do not make this claim if the reviewer could specifically point out where our analysis leads to the claim.

---

> > > ### Author Response · Authors · 2025-11-19
> > >
> > > > It is important to note that unifying various algorithms into a unique framework does not necessarily meet the standard for publication at a top ML conference. While the authors provide some discussion on how the unification will help future research (Sec. 6), I think the main points are mostly speculative.
> > >
> > > Please refer to our discussion about the benefits of our unification above. Our work provides a deeper theoretical understanding about the commonly used Unsupervised RL algorithms, drawing connections to a unified objective, with a number of implications as discussed in Section 6. We also illustrate how this unified perspective can lead to development of newer algorithms (some of them already demonstrated in our experiments). We have also
> > >
> > > We would like to point out that similar works [1, 2, 3] have been well received in prior ML conferences with [1] being an oral presentation at a prior ICLR conference. These works have provided a deeper understanding about the underlying fields and have led to newer algorithms.
> > >
> > > > Other than showing that a bunch of different methods can be understood as approximating a successor measure, it would be interesting to see whether it can be proved that any reward-free algorithm that improves the efficiency of policy inference with reward by a certain amount has to implicitly learn successor measures. See the paper Richens et al., General agents need world models, 2025 for potential inspiration.
> > >
> > > The unification aims to highlight the very point. We show that methods covering the largest classes of downstream rewards $R$ and being efficient in policy optimization (zero-shot methods) actually compute successor measures in the most complete way. In Section 3.1 we have highlighted the reasons to why efficient URL algorithms will require to compute successor measures implicitly or explicitly. As to proving that any method that improves the efficiency of policy inference *will have to implicitly learn successor measures*, we believe it is not always true. There are methods [4, 5] that use purely computer vision based techniques (ignoring the sequential nature of data) for representation learning and improve the efficiency of policy inference but do not learn anything related to successor measures.
> > >
> > >
> > > [1]: Eysenbach, Benjamin, Ruslan Salakhutdinov, and Sergey Levine. "The Information Geometry of Unsupervised Reinforcement Learning." International Conference on Learning Representations
> > >
> > > [2]: Rakelly, Kate, et al. "Which Mutual-Information Representation Learning Objectives are Sufficient for Control?." Advances in Neural Information Processing Systems 34 (2021): 26345-26357
> > >
> > > [3]:Alshammari, Shaden Naif, et al. "I-Con: A Unifying Framework for Representation Learning." The Thirteenth International Conference on Learning Representations.
> > >
> > > [4]:Rutav M Shah and Vikash Kumar. Rrl: Resnet as representation for reinforcement learning. In
> > > International Conference on Machine Learning
> > >
> > > [5]: Majumdar, Arjun et. al., Where are we in the search for an artificial visual cortex for embodied intelligence? Advances in Neural Information Processing Systems, volume 36

---

> > > > ### Comment · Reviewer_mkCJ · 2025-11-24
> > > >
> > > > I want to thank the authors for their very extensive replies. Unfortunately, I do not see relevant new information to mitigate my major concerns **What does unsupervised RL means** and **Main hypothesis** and I am planning to keep my score.
> > > >
> > > > As a minor note:
> > > >
> > > > > The work (Liu and Abbeel, Behavior From the Void: Unsupervised Active Pre-Training, 2021) not only explores but learns a representation that is then used by the Reward-Free phase. So this work is also solving the URL problem that we have described
> > > >
> > > > Sure, of course APT learns a state representation, like any other deep reinforcement learning algorithm. However, not all the state representations are successor measures.

---

> > > > > ### Author Response · Authors · 2025-11-28
> > > > >
> > > > > We appreciate the reviewer’s response. We believe that our rebuttal provided clarifications regarding the reviewer’s key concerns—particularly those on unsupervised RL and the main hypothesis—and we would be happy to elaborate further if the reviewer requires specific clarifications. We briefly reiterate our responses for these two concerns:
> > > > >
> > > > > **Main Hypothesis**: We have been clear in the paper (lines 020-022, 065-074, 131-133, Appendix D) that we are unifying the URL algorithms that learn quantities or structures about the environment that predict future outcomes. We have also pointed out that there are unsupervised RL algorithms (such as APT) that do not learn such quantities and are outside the scope of the unification (we have added a discussion in Appendix D about the scope of the unification). We have made changes to the introduction to be more clear about our hypothesis.
> > > > >
> > > > > **Unsupervised RL**: We clarify the contribution of the paper is to provide a unifying thread to the majority of URL algorithms for which we use the descriptions of unsupervised RL in prior work [1, 2, 3]. We provide a description that is general enough to include most of the algorithms that were proposed as unsupervised RL algorithms, rather than prescribe a definition of the field.
> > > > >
> > > > > [1] Laskin, Michael, et al. "URLB: Unsupervised Reinforcement Learning Benchmark." Deep RL Workshop NeurIPS 2021.
> > > > >
> > > > > [2] Touati, Ahmed, Jérémy Rapin, and Yann Ollivier. "Does Zero-Shot Reinforcement Learning Exist?." The Eleventh International Conference on Learning Representations.
> > > > >
> > > > > [3] Park, Seohong, Oleh Rybkin, and Sergey Levine. "METRA: Scalable Unsupervised RL with Metric-Aware Abstraction." The Twelfth International Conference on Learning Representations.

---

### Official Review · Reviewer_kSjZ · 2025-11-01

**Soundness:** 2
**Presentation:** 2
**Contribution:** 2
**Rating:** 4
**Confidence:** 3

**Summary:**

The paper proposes a unified framework for diverse unsupervised RL (URL) algorithms. The framework concerning Reward-Free Phase and Reward-based Phase is used to construct this unified framework. Within this framework, the core concept is the successor measure M_\pi(s,a,s+).

To make it tractable, different URL algorithm families define parametric approximation of the policy class using different latent representations. The authors give a thorough theoretical explanation about all these diverse URL algorithms.

I appreciate the authors can list the table to make the paper clear.

**Strengths:**

1. The authors use a unified framework to combine seemingly different unsupervised RL algorithms, which I think is a very novel topic.
2. All the assumptions and theoretical process are clarified (but I do not carefully check the correctness of the theoretical process).
3. The table and figure are clear.

**Weaknesses:**

1. Unfortunately, I feel that the author's integration of various methods comes across as somewhat forced. This is because the author imposes additional assumptions that are not originally required by the URL method. These assumptions are necessary to fit the method into the framework, but when they are not met, the method can no longer be incorporated. Therefore, I believe further analysis is needed to determine when these assumptions are violated.
2. When all assumptions are satisfied, these methods are incorporated into the framework. However, the author fails to further explore the applicability of these methods, that says, when a certain method should be used and the distinct focus of different methods. I believe such discussion is necessary to inspire the development of new algorithms based on this framework.
3. The citation on line 977 of Appendix B is missing.

Since I have not thoroughly checked the theory in detail and find the aforementioned weaknesses to be quite evident, I am inclined to assign a borderline reject score at this stage, with relatively low confidence.

**Questions:**

See weakness above

---

> ### Author Response · Authors · 2025-11-19
>
> We thank the reviewer for the assessment. We address the concerns raised by the reviewer below:
>
> > Unfortunately, I feel that the author's integration of various methods comes across as somewhat forced. This is because the author imposes additional assumptions that are not originally required by the URL method. These assumptions are necessary to fit the method into the framework, but when they are not met, the method can no longer be incorporated. Therefore, I believe further analysis is needed to determine when these assumptions are violated.
>
> We would like to clarify that these assumptions are not imposed by us on these respective paradigms to fit them into the unified framework. Prior research has imposed these unstated assumptions to produce tractable algorithms and  algorithmic paradigms. Our work makes this explicit and formal (in terms of reward classes and policy classes) and to provide a clear understanding of what assumptions are for these different paradigms. These assumptions have been mentioned in several prior papers as properties of these paradigms: GCRL[1], MISL[2,7], SF[3], PSM[6], PVF[4, 5], Controllable Representations[8] and world models [9].
> Changing these assumptions will lead to a new paradigm which is a takeaway of the unification as described in Section 6.
>
> > When all assumptions are satisfied, these methods are incorporated into the framework. However, the author fails to further explore the applicability of these methods, that says, when a certain method should be used and the distinct focus of different methods. I believe such discussion is necessary to inspire the development of new algorithms based on this framework.
>
> Table 2 describes the pros and cons of each method as a consequence of these assumptions. These give insights about the applicability of these methods. We would like to clarify that we do not talk about one method being better than all others as there is no free lunch. The assumptions and approximations made by each paradigm towards solving the unified framework formally describe the distinct focus and the strengths and weaknesses of each paradigm. Our work provides a theoretical foundation for a number of unsupervised RL algorithms and this clarification can have a number of implications; new algorithms, analysis of all methods under similar setup, and clarity on what methods will be applicable because we know the assumptions the methods are based on. Similar works have led to improved understanding in the field [2,7]. Section 6 clearly highlights how these discussions can lead to developing newer approaches. We provide some examples of using the best of two approaches. We produce additional examples in the revised version. An extensive empirical validation is beyond the scope of this work.
>
> > The citation on line 977 of Appendix B is missing.
>
> Thank you for pointing this out, we have corrected it.
>
>
> [1]: Leslie Pack Kaelbling. Learning to achieve goals. In IJCAI, 1993.
>
> [2]: Eysenbach, Benjamin, Ruslan Salakhutdinov, and Sergey Levine. "The Information Geometry of Unsupervised Reinforcement Learning." International Conference on Learning Representations
>
> [3]: Touati, Ahmed, Jérémy Rapin, and Yann Ollivier. "Does Zero-Shot Reinforcement Learning Exist?." The Eleventh International Conference on Learning Representations.
>
> [4]: Farebrother, Jesse, et al. "Proto-Value Networks: Scaling Representation Learning with Auxiliary Tasks." The Eleventh International Conference on Learning Representations
>
> [5]: Dadashi, Robert, et al. "The value function polytope in reinforcement learning." International Conference on Machine Learning. PMLR, 2019.
>
> [6]: Agarwal, Siddhant, et al. "Proto Successor Measure: Representing the Behavior Space of an RL Agent." Forty-second International Conference on Machine Learning.
>
> [7]: Rakelly, Kate, et al. "Which Mutual-Information Representation Learning Objectives are Sufficient for Control?." Advances in Neural Information Processing Systems 34 (2021): 26345-26357
>
> [8]: Levine, Alexander, Peter Stone, and Amy Zhang. "Multistep Inverse Is Not All You Need." Reinforcement Learning Conference.
>
> [9]: Ding, Zihan, et al. "Diffusion world model: Future modeling beyond step-by-step rollout for offline reinforcement learning." arXiv preprint arXiv:2402.03570 (2024).

---

> > ### Comment · Reviewer_kSjZ · 2025-11-20
> > **Response by Reviewer kSjZ**
> >
> > Thanks for the authors' clarification. The paper presents a commendable unified framework. However, I feel the potential is not fully realized. It would strengthen the paper if the authors could develop a novel algorithm based on this framework, thereby demonstrating the utility and practical value of this framework. Nevertheless, based on the authors' clarification, I find the paper's direction to be promising.
> >
> > I will adjust my score during the AC-Reviewer discussion.

---

### Official Review · Reviewer_aYbj · 2025-11-02

**Soundness:** 2
**Presentation:** 2
**Contribution:** 1
**Rating:** 2
**Confidence:** 3

**Summary:**

The authors propose a unifying framework for unsupervised reinforcement learning (URL) algorithms, arguing that seemingly disparate methods (including Goal-Conditioned RL, Mutual Information Skill Learning, Successor Features, Proto-Successor Measures, Proto-Value Functions, Controllable Representations, and World Models) can be viewed as approximating a common successor measure objective under different assumptions.

**Strengths:**

- The core idea of unifying disparate URL methods through the lens of successor measures is interesting and could provide value to the community's understanding of these algorithms.
- The paper covers seven algorithm families and derives connections between each and the proposed unified objective.

**Weaknesses:**

- While I think the idea of a unifying framework for Unsupervised RL relevant and interesting, I think the paper is incomplete:
    - the paper presents a numerous amount of methods and how they relate to the unified objective, but it should also justify why such unification is useful in the first place. In prior work [1], this has been done by finding and mixing synergies from different domains and showing that they can improve one another. Here this is quickly done with a very limited amount of tasks.
    - More analysis should be expected. In toy environments, I would expect to see how the different methods approximate $M^\pi$ in the reward-free phase: which method explores which part of the space, and why? How does this affect the reward-based phase?
- I do not understand the presence of a World Models paragraph. World models are not unsupervised RL algorithms. They're just models trained on a dataset, and the question is: how do you build the dataset, which will be used to better approximate $M^\pi$ (with the world model)? For getting diverse data, we can use unsupervised RL algorithms, such as RL algorithms with prediction disagreement rewards [2].
- Also, Figure 1 is difficult to read, and there should be at least some info about the tasks in the core paper (readers should not need to go at the end of appendix to find the task descriptions).


Minor:
- missing citations in first paragraph of 4.2.
- Section 4.3, I don't understand why using the vector reward here $\textbf{r}$ and the feature matrix $\Phi$ without introducing them. Why not simply using the simple formula $r=\phi^T w$ ?
- "While MISL approaches have large variation in their overall algorithms, the core has always been to maximize the mutual information between states and “skills” ($I(S, Z)$) or between transitions and skills ($I(S, S′; Z)$)" I would suggest soften the claim a bit, as there are other objectives like $I(S',Z ; S)$ [3] and $I((S,S'),Z)$ [4]
- there's a formatting issue at line 1452

[1] Choi, Jongwook, et al. "Variational empowerment as representation learning for goal-based reinforcement learning."
[2] Sekar, Ramanan, et al. "Planning to explore via self-supervised world models."
[3] Sharma, Archit, et al. "Dynamics-aware unsupervised discovery of skills."
[4] Laskin, Michael, et al. "Cic: Contrastive intrinsic control for unsupervised skill discovery."

**Questions:**

- Did you notice any other relevant synergies between the described methods?
- I can see that there are two kinds of tasks described in appendix E, which can be randomized, how many of these environments did you generate to evaluate the performance of each algorithm?

---

> ### Author Response · Authors · 2025-11-19
>
> We thank the reviewer for the review. We address the concerns raised by the reviewer below:
>
> > the paper presents a numerous amount of methods and how they relate to the unified objective, but it should also justify why such unification is useful in the first place. In prior work [1], this has been done by finding and mixing synergies from different domains and showing that they can improve one another. Here this is quickly done with a very limited amount of tasks.
>
> We justify the unification in Section 6. We provide experiments only as support for our existing arguments, which are that the unification provides theoretical clarity, clear stratification for novel algorithms and characterization of algorithmic categories. We are running experiments on additional mixed synergies,  more task distributions and continuous domains. We will update them in the overleaf and inform you once they are done.
> We are running experiments with more task distributions, more cross combinations of different methods and also on more continuous domains. We will inform you once we have updated them in the paper.
>
> > More analysis should be expected. In toy environments, I would expect to see how the different methods approximate M^\pi in the reward-free phase: which method explores which part of the space, and why? How does this affect the reward-based phase?
>
> As pointed out in Appendix E, all the algorithms are evaluated in an offline setting (no online exploration) with access to the entire space of transitions.. We are adding visualizations for successor measures learnt by different methods for some policies within their policy classes. We will update you when these visualizations are in the paper.
> > I do not understand the presence of a World Models paragraph. World models are not unsupervised RL algorithms. They're just models trained on a dataset, and the question is: how do you build the dataset, which will be used to better approximate M^\pi (with the world model)?
>
> We have modified the section on “World Models” to “Planning with World Models”to provide more clarity. We would like to clarify that world models also classify as representations learnt from the MDP using reward-free interactions. This representation has been used to optimize policies for a distribution of tasks using planning. We also show in Section 4.7 that successor measures and world models are not very different.
>
> > For getting diverse data, we can use unsupervised RL algorithms, such as RL algorithms with prediction disagreement rewards.
>
> There are a lot of algorithms that do not use rewards (e.g. imitation learning methods, preference learning methods) and all do not fall into the definition of unsupervised RL. Exploration methods are one such class of approaches. We provide a clean definition of unsupervised RL which in agreement with prior literature [1].  According to the URL definition, the algorithm needs to have both, an objective to abstract knowledge from reward-free interactions into any form of representation and an objective to efficiently extract policies from the representation. While exploration using various methods such as prediction disagreement use reward-free objectives they do not entirely qualify as  URL algorithms, combining them with a method that learns a representation on the dataset collected that can actually improve the efficiency of policy optimization will make them URL algorithms (as done in [2]). We have added a discussion about exploration methods and the scope of our unification in Appendix C and D.
>
> > Also, Figure 1 is difficult to read, and there should be at least some info about the tasks in the core paper (readers should not need to go at the end of appendix to find the task descriptions)’
>
> Thank you for pointing it out. We have cleaned the plot in the revised version. The goal of the paper was to present this unification so experiment details were moved to the Appendix. We have moved some experiment details in the main paper.
>
> > missing citations in first paragraph of 4.2.
>
> Thank you for pointing out, we have corrected this.
>
> > Section 4.3, I don't understand why using the vector reward here r and the feature matrix \Phi without introducing them. Why not simply using the simple formula r=\phi^Tw ?
>
> We tried to keep the definition of rewards as close to the definitions used in the respective work. We have moved to a functional definition: $r(s) = \sum_i \phi_i(s) w_i$.

---

> ### Author Response · Authors · 2025-11-19
>
> > "While MISL approaches have large variation in their overall algorithms, the core has always been to maximize the mutual information between states and “skills” (I(S, Z)) or between transitions and skills (I(S,S’;Z))" I would suggest soften the claim a bit, as there are other objectives like  I(S’,S;Z)[3] and I((S,S’),Z) [4]
>
> The mutual information objectives that we mention are $I(S;Z)$ (MI between the states and skills) and $I(S,S’;Z)$ (MI between transitions and skills). These mutual information are often implicitly conditioned on initial state similar to [3] ($I(S’;Z|S)$ ) This is how we get the variational distribution $q(z|s^+, s)$ in Theorem 4.6. Similarly [4] uses exactly the same definition as $I(S,S’;Z)$ i.e. MI between transitions and skills. We agree that the mutual information objectives have been described in a number of ways but they are largely seen as one of the two objectives $I(S;Z)$ and $I(S,S’;Z)$ [5].
>
> > there's a formatting issue at line 1452
>
> Thank you for pointing out, we have corrected this.
>
> > Did you notice any other relevant synergies between the described methods?
>
> Through the unification we observed that all methods are approximating successor measures during the Reward-Free phase and using the successor measure in the Reward-Based phase. This allowed us to create a few cross-combinations (some of which have been studied in prior literature). We believe that this unified perspective provides a holistic way to study these algorithms and create more algorithms using the objectives and tricks for different algorithm paradigms.
>
> > I can see that there are two kinds of tasks described in appendix E, which can be randomized, how many of these environments did you generate to evaluate the performance of each algorithm?
>
> We used 10 random samples of each set of tasks averaged across 4 seeds.
>
> [1]: Laskin, Michael, et al. "URLB: Unsupervised Reinforcement Learning Benchmark." Deep RL Workshop NeurIPS 2021.
> [2]: Liu and Abbeel, “Behavior From the Void: Unsupervised Active Pre-Training”, 2021
> [3]: Sharma, Archit, et al. "Dynamics-aware unsupervised discovery of skills.”
> [4]: Laskin, Michael, et al. "Cic: Contrastive intrinsic control for unsupervised skill discovery."
> [5]: Zheng, Chongyi, et al. "Can a MISL Fly? Analysis and Ingredients for Mutual Information Skill Learning." The Thirteenth International Conference on Learning Representations.

---

### Official Review · Reviewer_pWhB · 2025-11-03

**Soundness:** 3
**Presentation:** 3
**Contribution:** 3
**Rating:** 6
**Confidence:** 3

**Summary:**

The paper tackles unsupervised / reward-free reinforcement learning (URL) as a pretraining paradigm that learns task-agnostic structure which can later be reused for efficient downstream policy inference once a reward is specified. The authors argue that several prominent URL families: Goal-Conditioned RL (GCRL), Mutual-Information Skill Learning (MISL), Successor Features (SF), Proto-Successor Measures (PSM), Proto-Value Functions (PVF), Controllable Representations (CR), and World Models, can be understood through a single unifying theory: they each (explicitly or implicitly) learn approximations to a successor measure $M^\pi$, i.e., the discounted measure over future states visited when starting from  $(s,a)$ and following policy $\pi$. Under this view, downstream policy optimization for any reward becomes linear in $M^\pi$, which is an attempt to clarify why URL pretraining can enable fast and efficient policy inference across many tasks. The paper formalizes this perspective, frames each family as making tractable approximations to a conceptually intractable unified objective, and highlights design trade-offs between expressivity (size of the represented policy class) and inference efficiency (cost of search once rewards are given).

**Strengths:**

To support the conceptual claims, the paper presents a four-rooms gridworld study with a broad task distribution. The key result (Fig. 1) visualizes a trade-off between downstream performance and reward-phase training cost across URL families, consistent with the unifying theory’s expressivity–efficiency view. The paper offers an interesting perspective but there are some observations and questions.

**Weaknesses:**

- The theoretical unification is novel and timely; it convincingly reframes disparate URL families within a single successor-measure lens. To further guide readers, I recommend adding a concise schematic taxonomy that (i) situates URL within the broader RL landscape, and (ii) maps each covered family, GCRL, MISL, SF/PSM/PVF, Controllable Representations, World Models, to the unified view. A figure could show: a top level split (standard reward-based RL vs. reward-free pretraining/URL), within URL, nodes per family annotated with its learned representation RRR, the policy class it supports, and the induced state-abstraction metric. edges/arrows indicating how each family is a tractable approximation to the intractable unified objective, among others. A companion “cross-walk” table aligning each family’s objective with its $d$, $Π$, and how it leverages successor measures would make the unification operational and substantially improve reader orientation.

- If one wished to design a new URL method within the authors’ unified framework, the natural control knob is the choice of similarity/metric $d$ that induces the state abstraction $\phi$. Making this design space explicit would turn the framework from descriptive to prescriptive and guide practitioners on how to instantiate new methods. Concretely, I suggest: (i) add a small theorem/assumption block stating minimal conditions on $d$ to guarantee the successor-measure–based state equivalence via $\phi$ and (ii) include a small sensitivity study that swaps 𝑑 with some functions like: cosine, RBF kernel,  energy score, among others, in your gridworld to show how changes the expressivity and inference-efficiency trade-off and the induced abstractions.

- The paper unifies several major URL families, but influential directions such as curiosity-driven exploration, empowerment-based methods, and contrastive predictive control are not explicitly discussed. A short clarification on how these paradigms would fit within the successor-measure view (or whether they fall outside its scope) would strengthen the completeness of the framework and help position its boundaries.

- Minor stylistic note: the section title “Consequences of the Unification” reads slightly ambiguous, as “consequences” can carry a negative connotation in some linguistic and academic contexts. A more affirmative phrasing such as “Implications of the Unification”, or “Theoretical and Practical Implications” might highlight the constructive nature of the results. Although this is a very minor stylistic comment.

**Questions:**

- Table 1 introduces a method-dependent similarity $d$ to induce state-abstraction/equivalence, and the text notes that different URL families instantiate $d$  via norms, dot-products, or KL-divergences (thereby operationalizing $\phi$  and the successor-measure view). This is a powerful unifying hook, but it naturally raises design-space questions the paper could address more explicitly. (a) What happens if we change $d$? beyond the listed instances, could cosine, kernel/RBF, or energy-based similarities yield new algorithmic families (rather than retrofitting existing ones)? (b) Which assumptions must $d$ satisfy to preserve the successor-measure equivalence that underlies the abstraction. Some guidance linked to Def. 5.1 and your $\phi$-based equivalence would make the framework more actionable.

---

> ### Author Response · Authors · 2025-11-19
>
> We thank the reviewer for their assessment. We address the concerns raised as follows:
>
> > The theoretical unification is novel and timely; it convincingly reframes disparate URL families within a single successor-measure lens. To further guide readers, I recommend adding a concise schematic taxonomy that (i) situates URL within the broader RL landscape, and (ii) maps each covered family, GCRL, MISL, SF/PSM/PVF, Controllable Representations, World Models, to the unified view. A figure could show: a top level split (standard reward-based RL vs. reward-free pretraining/URL), within URL, nodes per family annotated with its learned representation RRR, the policy class it supports, and the induced state-abstraction metric. edges/arrows indicating how each family is a tractable approximation to the intractable unified objective, among others. A companion “cross-walk” table aligning each family’s objective with its d, \Pi, and how it leverages successor measures would make the unification operational and substantially improve reader orientation.
>
> Thank you for your constructive suggestion. We agree that this could be a very useful addition to the paper. We are working on it and will share with you once ready.
>
> > If one wished to design a new URL method within the authors’ unified framework, the natural control knob is the choice of similarity/metric d that induces the state abstraction \phi. Making this design space explicit would turn the framework from descriptive to prescriptive and guide practitioners on how to instantiate new methods. Concretely, I suggest: (i) add a small theorem/assumption block stating minimal conditions on  to guarantee the successor-measure–based state equivalence via d and (ii) include a small sensitivity study that swaps 𝑑 with some functions like: cosine, RBF kernel, energy score, among others, in your gridworld to show how changes the expressivity and inference-efficiency trade-off and the induced abstractions.
>
> While $d$ is an important factor in designing the URL algorithm, a lot of algorithm families (eg. SF, PVF) do not explicitly use $d$ to construct their algorithms. The distance metric among the state compressions comes implicitly due to their objectives. Changing the metric here can lead to significant changes in the algorithm, in some cases changing the algorithmic paradigm as well. An example of this could be: [1] and [2] have very similar losses (Contrastive losses) while [1] is a goal conditioned RL algorithm and [2] is a representative work for successor features. Through this paper, we highlight these connections among different approaches that they are solving the same objective with different parameterizations and assumptions. Hence the experiments that we have in the paper that compare these different approaches, actually compare the different metrics $d$ used. In our experiments, we have made as much comparison as makes sense without significantly changing the algorithm.
>
> The choice of $d$ is a structure imposed in the representation space of $\phi$. The Definition 5.1 and the discussion following it simply says that two states are similar if the successor measures induced by the two states (distribution of futures induced by the states) are similar. For two states that are similar, their representations also need to be close to each other according to the metric space of the representation space of $\phi$. To put it simply, the choice of $d$ puts an inductive bias on the representation space and can be independent from the underlying MDP. We have added some discussion in Section 5 highlighting this.

---

> ### Author Response · Authors · 2025-11-19
>
> > The paper unifies several major URL families, but influential directions such as curiosity-driven exploration, empowerment-based methods, and contrastive predictive control are not explicitly discussed. A short clarification on how these paradigms would fit within the successor-measure view (or whether they fall outside its scope) would strengthen the completeness of the framework and help position its boundaries.
>
> We add a brief discussion about these methods in the Appendix B.2, C, E.1. In short, the empowerment methods are closely related to MISL as pointed out by [4], where the empowerment objective is the same as MISL objective with the skill parameter $z$ being used to parameterizing the action sequences. Contrastive Predictive Methods (like the ones using InfoNCE) [1,3] have been shown to be performing GCRL[1] and will fall under the same paradigm.
> We would like to point out that the exploration methods do not fall into the URL problem definition in Section 2 as they do not provide any mechanism to extract optimal policies. Since the goal of URL algorithms are to improve the efficiency of policy extraction for any reward, the algorithms should consist of both, a representation learning objective (representation could be anything ranging from policies and value functions to transition models) and the corresponding policy extraction objective. We discuss more about exploration and their connection to our unified perspective in Appendix C.
>
> > Minor stylistic note: the section title “Consequences of the Unification” reads slightly ambiguous, as “consequences” can carry a negative connotation in some linguistic and academic contexts. A more affirmative phrasing such as “Implications of the Unification”, or “Theoretical and Practical Implications” might highlight the constructive nature of the results. Although this is a very minor stylistic comment.
>
> We thank the reviewer for the suggestion. We have modified the section title in the revised version.
>
> > Table 1 introduces a method-dependent similarity d to induce state-abstraction/equivalence, and the text notes that different URL families instantiate d via norms, dot-products, or KL-divergences (thereby operationalizing  and the successor-measure view). This is a powerful unifying hook, but it naturally raises design-space questions the paper could address more explicitly. (a) What happens if we change d?
>
> beyond the listed instances, could cosine, kernel/RBF, or energy-based similarities yield new algorithmic families (rather than retrofitting existing ones)? (b) Which assumptions must  satisfy to preserve the successor-measure equivalence that underlies the abstraction. Some guidance linked to Def. 5.1 and your -based equivalence would make the framework more actionable.
>
> Cosine will be very similar to the inner product based similarity distance with constant $||\phi||$. Using other distance formalisms can lead to different algorithms. In principle, you can always parameterize the different quantities using different parameterizations as per the choice of distance metrics and run the algorithm paradigms. The study of the overall structure imposed by the choice of distance metric on the representation space leading to a different set of policy class $\Pi$ and reward class $\mathcal{T}$ is interesting and is a potential direction for future research.
>
> There aren't any assumptions about $d$ apart from the fact that $d(\phi(s_1), \phi(s_2)) \propto p(s_1 = s_2)$. So, it depends on what structure is required in the representation space. If you desire Euclidean structure, using L2 norm should be the choice for $d$; if you require a spherical or spectral structure, you might want to go with an inner product based similarity distance. If the choice of the space is some non-Euclidean manifold, the choice for $d$ will be according to the manifold.
>
> [1]: Eysenbach, Benjamin, et al. "Contrastive learning as goal-conditioned reinforcement learning." Advances in Neural Information Processing Systems 35 (2022): 35603-35620.
> [2]: Touati, Ahmed, and Yann Ollivier. "Learning one representation to optimize all rewards." Advances in Neural Information Processing Systems 34 (2021): 13-23.
> [3]: Ziarko, Alicja, et al. "Contrastive Representations for Temporal Reasoning." The Thirty-ninth Annual Conference on Neural Information Processing Systems.
> [4]: Eysenbach, Benjamin, et. al. "Diversity is all you need: Learning skills without a reward function". 2018

---

### Author Response · Authors · 2025-12-03
**General Response**

We thank the reviewers for the constructive feedback. The paper introduces a unification of a substantial number  of unsupervised RL algorithms through the common thread that these algorithms learn models about predicting future outcomes in the environment. In particular, we show that the algorithmic paradigms: Goal Conditioned RL, Successor Features, Mutual Information Skill Learning, Proto Value Functions, Proto Successor Measures, World Models and Controllable Representations, are approximations of a unified, intractable objective. Through this unification we:

- Provide a common ground to study the strengths and weaknesses of each algorithm through approximations and assumptions made by them towards solving the unified unsupervised RL problem.

- Show through these approximations, that no algorithm is the best in all situations. Experimentally, we produce a pareto frontier showing the performance vs inference cost for these algorithm families.

- We provide a framework for novel algorithm design by cross combination of different algorithm families. We also present novel algorithms that can easily be constructed using our framework.

All reviewers have acknowledged that **our work is novel, timely, clear and covers a wide variety of disparate URL algorithms**. None of the reviewers contest the above contributions of our work which we justify both theoretically and empirically. Furthermore, Reviewers **aYbj** and **mkCJ** have agreed that **the community will benefit from the paper as it provides a deeper understanding about these algorithms**. We have provided substantive clarifications to the concerns raised by the reviewers. Some of the major ones are:

> “Simply showing unification is not enough”. Reviewers **aYbj**, **kSjZ** (in the response to our rebuttal) and **mkCJ** have argued that showing unification is not enough. Reviewers **aYbj** and **KSjZ** have suggested development of new algorithms using the unification and Reviewer **mkCJ** claimed that the unification does not meet the standards for a publication in a top ML conference.

However, there has been prior work [1, 2] that has been well received by the community and top ML conferences ([1] being an oral in a prior ICLR) for providing theoretical insights on existing algorithms without introducing novel algorithms. The insights in these prior works inspired future algorithms, which we believe will be the contribution of our unification. Through the lens of unification, our work provides a theoretical foundation for a number of unsupervised RL algorithms that improve the understanding of conceptual frameworks and clarify the strengths and weaknesses of these approaches.  We have also highlighted (corroborated by both our experiments and existing algorithms) how these observations can lead to novel algorithms by combining principles from the two paradigms, and illustrate some of these novel algorithms in our rebuttal. In particular, we have added more algorithms produced using cross combinations of different algorithm families and also included a different policy inference method which improves performance of several algorithm families at some cost of increased computation.

> Reviewer **mkCJ** argues that our description of “Unsupervised RL” is not clear enough.

We clarify the contribution of the paper is to provide a unifying thread to a significant number of algorithms that are described as URL according to the descriptions of unsupervised RL in prior work [3, 4, 5]. Our objective in this work is **not to prescribe what unsupervised RL is**. We are instead  **describing a unifying thread between algorithms** that already purport to be URL The objective of the paper has been to use this description to cover as many of the self-described unsupervised RL algorithms as possible. We describe the goal of URL as being to amortize the computation of RL algorithms into two phases, leading to improved efficiency of the Reward-Based phase. This improvement depends on how successful an underlying algorithm has been at amortizing the computation. An algorithm that does not improve the efficiency significantly could still be described as a URL algorithm—just arguably not a good one.

> Reviewer **mkCJ** believes that we are making a strong claim by “unifying URL”.

We have been clear in our paper (lines 020-022, 065-074, 131-133, Appendix D) that we are not unifying **all** unsupervised RL algorithms rather unifying a significant number of methods with the common thread of modeling future outcomes. There are algorithms [6, 7] which are unsupervised RL algorithms but do not fall in scope of the unification. We have added a separate discussion about the scope of our unification in Appendix D.

---

> ### Author Response · Authors · 2025-12-03
>
> We have made other modifications to our paper to address some of the concerns:
>
> - We have added a discussion on exploration methods with their connection to our unification in Appendix C. (suggested by Reviewers **pWhB**, **aYbj** and **EHY5**)
>
> - We have added some Experiment details to the main paper and updated Figure depicting the experiments (as suggested by Reviewer **aYbj**).
>
> - We have added a main Figure to show an overview of the unification (as suggested by Reviewer **pWhB**).
>
> - We demonstrate the design of novel algorithms by using various objectives for the Reward-Free phase that approximate successor measures and combining them with various Reward-Based objectives.
>
> - We visualize the successor measures learnt by different methods.
>
> [1]: Eysenbach, Benjamin, Ruslan Salakhutdinov, and Sergey Levine. "The Information Geometry of Unsupervised Reinforcement Learning." International Conference on Learning Representations
>
> [2]: Rakelly, Kate, et al. "Which Mutual-Information Representation Learning Objectives are Sufficient for Control?." Advances in Neural Information Processing Systems 34 (2021): 26345-26357
>
> [3] Laskin, Michael, et al. "URLB: Unsupervised Reinforcement Learning Benchmark." Deep RL Workshop NeurIPS 2021.
>
> [4] Touati, Ahmed, Jérémy Rapin, and Yann Ollivier. "Does Zero-Shot Reinforcement Learning Exist?." The Eleventh International Conference on Learning Representations.
>
> [5] Park, Seohong, Oleh Rybkin, and Sergey Levine. "METRA: Scalable Unsupervised RL with Metric-Aware Abstraction." The Twelfth International Conference on Learning Representations.
>
> [6]:Rutav M Shah and Vikash Kumar. Rrl: Resnet as representation for reinforcement learning. In International Conference on Machine Learning
>
> [7]: Liu and Abbeel, Behavior From the Void: Unsupervised Active Pre-Training, 2021

---

### Meta-Review · Area_Chair_oEd5 · 2026-01-06

**Summary:**

This paper proposes a unified framework for unsupervised reinforcement learning (URL) algorithms, arguing that methods like Goal-Conditioned RL (GCRL), Mutual Information Skill Learning (MISL), Successor Features (SF), and World Models approximate a common successor measure objective under different assumptions. While reviewers acknowledge the novelty and timeliness of unifying disparate URL families, fundamental concerns persist: (1) the URL problem definition lacks sufficient rigor to support the theoretical claims [Reviewer mkCJ], (2) the practical value of the unification for guiding future algorithm development remains speculative with limited empirical support [Reviewers mkCJ, aYbj, kSjZ], and (3) the scope of "unifying URL" may be overstated given methods that fall outside the framework [Reviewer mkCJ].

**Reviewer Concerns:**

**Addressed by Rebuttal:**
- The authors clarified that exploration methods alone do not constitute URL algorithms: they must include both representation learning and policy extraction objectives [Reviewer aYbj, EHY5, pWhB].
- The authors added discussions on exploration methods (Appendix C) and scope of unification (Appendix D) to address concerns about framework boundaries [Reviewers pWhB, aYbj, EHY5].
- Missing citations and formatting issues were addressed [Reviewers aYbj, kSjZ].

**Outstanding Concerns:**
- The definition of unsupervised RL remains insufficiently rigorous, particularly regarding what constitutes "improved efficiency" in the reward-based phase [Reviewer mkCJ].
- The claim of "unifying URL" may be overstated given that notable methods like curiosity-driven exploration and some policy pre-training approaches fall outside the framework [Reviewers mkCJ, aYbj].
- The practical utility of the unification for developing new algorithms remains speculative, with limited empirical validation [Reviewers mkCJ, aYbj, kSjZ].
- The assumption that all unified methods implicitly learn successor measures is contested, methods like APT, learn state representations but not necessarily successor measures [Reviewer mkCJ].

**Reviewer Scores:**

Reviewer mkCJ: 2. The reviewer may maintain their score given that the major concerns regarding the URL problem definition and the main hypothesis about successor measures were not sufficiently addressed. The clarifications provided did not introduce new information to mitigate these fundamental issues.

Reviewer kSjZ: 4. The reviewer may slightly increase their score based on author clarifications about assumptions and the unified framework's utility, though concerns about forced integration of methods and lack of novel algorithm development persist.

Reviewer pWhB: 6. The reviewer may maintain their score as the paper's direction is promising and the theoretical unification is sound, though additional schematic taxonomy and design-space guidance would strengthen the contribution.

Reviewer aYbj: 2. The reviewer may maintain their score given that the justification for why unification is useful remains weak, the experimental validation is limited, and the presence of world models in the framework remains unclear.

Reviewer EHY5: 8. The reviewer may maintain their positive assessment as the authors adequately addressed concerns about entropy-based methods and GCRL variants, and the framework covers a wide range of URL algorithms with clear theoretical contributions.

---

### Decision · Program_Chairs · 2026-01-26

Reject